# An RNA-binding atypical tropomyosin recruits kinesin-1 dynamically to *oskar* mRNPs

Imre Gáspár[*] ID, Vasiliy Sysoev ID, Artem Komissarov ID & Anne Ephrussi[**] ID

## Abstract

Localization and local translation of *oskar* mRNA at the posterior pole of the *Drosophila* oocyte directs abdominal patterning and germline formation in the embryo. The process requires recruitment and precise regulation of motor proteins to form transport-competent mRNPs. We show that the posterior-targeting kinesin-1 is loaded upon nuclear export of *oskar* mRNPs, prior to their dynein-dependent transport from the nurse cells into the oocyte. We demonstrate that kinesin-1 recruitment requires the *Dm*Tropomyosin1-I/C isoform, an atypical RNA-binding tropomyosin that binds directly to dimerizing *oskar* 3′UTRs. Finally, we show that a small but dynamically changing subset of *oskar* mRNPs gets loaded with inactive kinesin-1 and that the motor is activated during mid-oogenesis by the functionalized spliced *oskar* RNA localization element. This inefficient, dynamic recruitment of Khc decoupled from cargo-dependent motor activation constitutes an optimized, coordinated mechanism of mRNP transport, by minimizing interference with other cargo-transport processes and between the cargo-associated dynein and kinesin-1.

**Keywords** active transport; atypical tropomyosin isoform; molecular motor; oocyte; RNA binding protein

**Subject Categories** Cell Adhesion, Polarity & Cytoskeleton; Development & Differentiation; Membrane & Intracellular Transport

The EMBO Journal (2017) 36: 319–333

## Introduction

Within cells, organelles, diverse macromolecules and complexes depend on a small set of cytoskeleton-associated motor proteins to achieve their proper distributions. Targeted delivery is ensured by cargo-associated guidance cues that are responsible for recruiting the appropriate mechanoenzyme (Hirokawa *et al*, 2010). Such actively transported cargoes include mRNAs whose asymmetric localization and local translation within cells is essential for various cellular functions, such as migration, maintenance of polarity and cell fate specification (Medioni *et al*, 2012). In the case of

messenger ribonucleoprotein (mRNP) particles, the guidance cues are the mRNA localization elements (LEs) that suffice to drive localization of the RNA molecule that contains them (Marchand *et al*, 2012). A few LEs, their RNA binding proteins (RBP) and the factors that link them to the mechanoenzyme have been well characterized (Dienstbier *et al*, 2009; Bullock *et al*, 2010; Dix *et al*, 2013; Niedner *et al*, 2014). In these cases, the entire localization process is driven by a single type of motor. Other mRNAs, such as *Xenopus laevis Vg1* (Gagnon *et al*, 2013) and *Drosophila melanogaster oskar* (Clark *et al*, 2007; Zimyanin *et al*, 2008; Jambor *et al*, 2014), rely on the coordinated action of multiple motor proteins—cytoplasmic dynein and kinesin-1 and kinesin-2 family members—for their localization within developing oocytes.

*oskar* mRNA encodes the posterior determinant Oskar protein, which induces abdomen and germline formation in a dosage-dependent manner in the fly embryo (Ephrussi & Lehmann, 1992). *oskar* mRNA is transcribed in the nurse cells of the germline syncytium and transported into the oocyte, similar to, for example, *bicoid* and *gurken* mRNAs (Ephrussi *et al*, 1991; Kim-Ha *et al*, 1991). This first step of *oskar* transport is guided by a well-described LE, the oocyte entry signal found in the 3′UTR of the mRNA (Jambor *et al*, 2014), which is thought to recruit the Egl–BicD–dynein transport machinery (Clark *et al*, 2007; Jambor *et al*, 2014). In the oocyte, *oskar* mRNA localization to the posterior pole is mediated by kinesin-1 (Brendza *et al*, 2000; Zimyanin *et al*, 2008; Loiseau *et al*, 2010). This second step of *oskar* transport requires splicing of the first intron in the *oskar* pre-mRNA, as *oskar* transcripts lacking all three introns (*oskar* Δi(1,2,3)) or just intron 1 (*oskar* Δi(1)) fail to localize (Hachet & Ephrussi, 2004). This splicing event results in assembly of the spliced localization element (SOLE) and deposition of the exon junction complex (EJC) on the mRNA (Ghosh *et al*, 2012). The EJC/SOLE constitutes a functional unit that is crucial for maintaining efficient kinesin-1-dependent transport of *oskar* mRNPs within the oocyte (Zimyanin *et al*, 2008; Ghosh *et al*, 2012), which is essential for proper localization of the mRNA to the posterior pole.

In a forward genetic screen, we identified a group of *DmTm1* (formerly *DmTmII*) mutants (*Tm1^{gs}*) in which *oskar* mRNA accumulation at the posterior pole of the oocyte fails (Erdelyi *et al*, 1995; Fig 1A and B). Although the small amount of Oskar protein produced at the posterior pole is sufficient for embryo progeny of *Tm1^{gs}* homozygous females to form an abdomen and develop into

Developmental Biology Unit, European Molecular Biology Laboratory, Heidelberg, Germany
*Corresponding author. Tel: +49 6221 387 8845; E-mail: gaspar@embl.de
**Corresponding author. Tel: +49 6221 387 8429; E-mail: ephrussi@embl.de

adult flies, it is insufficient to induce primordial germ cell formation. Consequently, the $Tm1^{gs}$ progeny are sterile, resulting in a so-called grandchildless phenotype (Erdelyi *et al*, 1995). It was subsequently demonstrated that the microtubule-mediated intra-ooplasmic motility of *oskar* mRNPs is affected in $Tm1^{gs}$ mutants (Zimyanin *et al*, 2008; Appendix Table S1), similar to what has been observed when kinesin-1 is absent (Zimyanin *et al*, 2008). Although there is biochemical evidence that kinesin-1 associates with *oskar* mRNPs (Sanghavi *et al*, 2013), what mediates the association of the motor with the mRNA, and where in the egg-chamber this occurs, is not known.

Here, we demonstrate that DmTm1-I/C, which consists mainly of low-complexity sequences and a C-terminal short tropomyosin superfamily domain (Cho *et al*, 2016), is an RNA-binding tropomyosin that recruits kinesin heavy chain (Khc) to *oskar* mRNA molecules upon their nuclear export in the nurse cells. We show that Khc recruitment to *oskar* RNPs is transient and dynamic and that this dynamic recruitment depends on the presence of DmTm1-I/C. Our data indicate that during mid-oogenesis, the EJC/SOLE triggers kinesin-1 activity, which drives localization of *oskar* mRNA to the posterior pole of the oocyte.

## Results

### Tm1-I/C maintains proper levels of kinesin-1 on *oskar* mRNA

To obtain mechanistic information regarding the motility defect in $Tm1^{gs}$ oocytes, we developed an *ex vivo* assay that allows co-visualization of MS2-tagged *oskar* mRNPs and polarity-marked microtubules (MTs) in ooplasm and determination of the directionality of *oskar* mRNP runs (Figs 1C and EV1A, and Video EV1), thus giving insight into the identity of the motor(s) affected by the $Tm1^{gs}$ mutations. Using this assay, we found that in wild-type ooplasm, plus-end-directed runs of *oskMS2* RNPs dominated about two to one over minus-end-directed runs (Fig 1D). Plus-end dominance was lost both in ooplasm lacking Khc and in extracts prepared from $Tm1^{gs}$-mutant oocytes (Fig 1D). This indicates that plus-end-directed, Khc-mediated motility is selectively compromised in the $Tm1^{gs}$ mutants. The remaining plus-end-directed runs might be due to residual kinesin-1 activity, to other plus-end-directed kinesins, or to cytoplasmic dynein, which has been shown to mediate the bidirectional random walks of mRNPs along MTs (Soundararajan & Bullock, 2014). To test whether the loss of Khc activity might be the cause of *oskar* mislocalization in $Tm1^{gs}$ oocytes, we tethered a minimal Khc motor, $Khc_{401}$ (Sung *et al*, 2008; Telley *et al*, 2009), to the MS2-tagged *oskar* mRNPs. Tethering of $Khc_{401}$–MCP to *oskMS2* restored the plus-end dominance of *oskar* mRNP runs (Fig 1D), as well as localization of *oskar* mRNA (Fig 1E and F), indicating that in $Tm1^{gs}$ mutants a loss of kinesin-1 activity might be the cause of *oskar* mislocalization.

The *DmTm1* locus encodes at least 17 different transcripts and 16 different polypeptides (Appendix Fig S1A). By performing semi-quantitative RT–PCR analysis, we found that the transcripts of Tm1-C, Tm1-I and Tm1-H are selectively missing or their amount is greatly reduced in $Tm1^{eg1}$ and $Tm1^{eg9}$ homozygous ovaries, respectively (Appendix Fig S1B and C). Furthermore, an EmGFP-Tm1-I

transgene expressed in the female germline rescued *oskar* mislocalization (Fig 1G) and the consequent grandchildless phenotype of $Tm1^{gs}$ mutants (all female progeny ($n > 20$) contained at least one ovary with developing egg-chambers). This indicates that the Tm1-I/C isoform is essential for *oskar* mRNA localization.

To determine whether the reduction in Khc-dependent *oskar* mRNP motility in $Tm1^{gs}$ oocytes is due to insufficient kinesin-1 recruitment or to insufficient motor activity, we analysed the composition of *oskar* mRNPs in ooplasms of flies co-expressing either Khc-EGFP or EmGFP-Tm1-I and *oskMS2*-mCherry, *ex vivo*

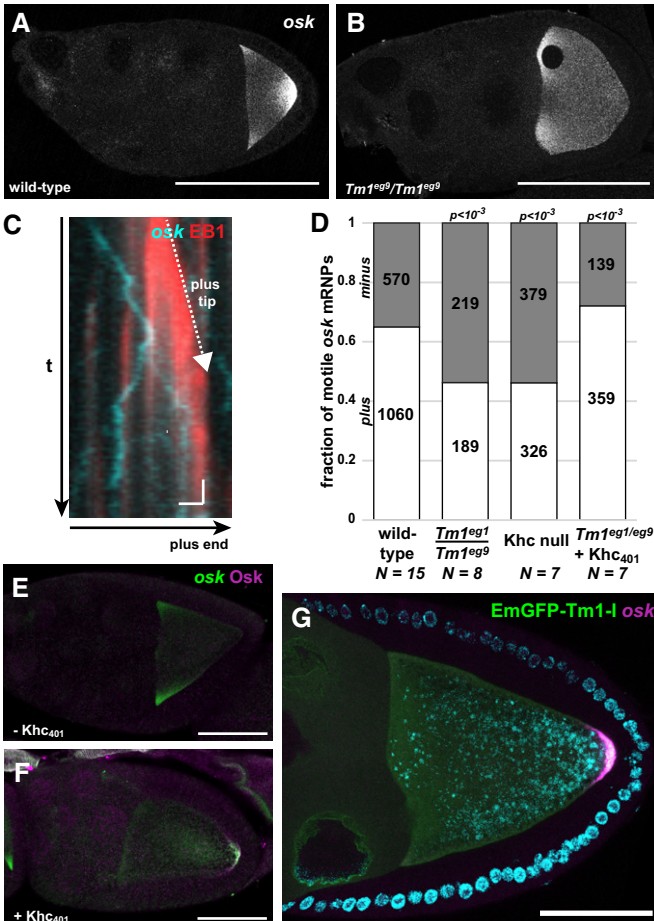

**Figure 1.  Effect of $Tm1^{gs}$ on *oskar* transport.**

A, B   Localization of *oskar* mRNA in wild-type (A) and $Tm1^{eg9}$/$Tm1^{eg9}$ (B) egg-chambers.

C   Kymograph of *oskMS2*-GFP mRNPs (cyan) travelling along polarity-marked MTs (red, EB1 protein) in an *ex vivo* ooplasmic preparation. White dashed arrow shows the growing plus tip of the MT. Scale bars represent 1 s and 1 μm, respectively.

D   Distribution of *oskMS2*-GFP mRNP runs towards plus (white) and minus ends (grey). Numbers within the bars indicate the number of runs. *P*-value of chi-squared test against wild type is indicated above each bar.

E, F   *oskar* mRNA (green) and Oskar protein (magenta) distribution in $Tm1^{eg1}$/$Tm1^{eg9}$ egg-chambers expressing *oskarMS2(6x)* (E) or *oskarMS2(6x)* and $Khc_{401}$–MCP (F).

G   *oskar* mRNA (magenta) distribution in $Tm1^{eg1}$/$Tm1^{eg9}$ egg-chambers rescued with EmGFP-Tm1-I (green).

Data information: Scale bars in (A, B and E–G) represent 50 μm.

(Videos EV2 and EV3). We first performed an object-based colocalization analysis of single snapshot images corrected for random colocalization (Fig EV1E–H). This revealed that both Khc-EGFP (Fig 2A and Video EV2) and EmGFP-Tm1-I (Fig 2B and Video EV3) are recruited to a small but significant fraction of *oskMS2*-mCherry mRNPs (1–4%; Fig 2C), indicating that the two proteins are components of *oskar* transport particles (Fig 2C). Furthermore, the association of Khc-EGFP with *oskMS2*-mCherry mRNPs was significantly reduced (two- to four-fold) in *Tm1$^{gs}$*-mutant ooplasms (Figs 2D and EV1H), indicating that the observed motility defects in *Tm1$^{gs}$*-mutant oocytes are due to insufficient kinesin-1 recruitment to *oskar* mRNPs.

In a complementary approach, we analysed Khc and Tm1-I colocalization with *oskMS2*-EGFP in consecutive images of entire time series. This assay estimates the probability of random colocalization in a different manner (Fig EV2A). For the analysis, we made use of flies expressing endogenously tagged Khc and Tm1-I/C, in which virtually all molecules of interest are labelled (Fig 3A and B; Appendix Fig S1C and G). This revealed that in *Khc$^{mKate2}$* homozygous ooplasmic extracts, nearly 50% of motile *oskMS2*-EGFP mRNPs are associated with Khc during at least half of the recorded trajectories (Fig 3C). As this value is close to the proportion of plus-end-directed runs (65%)—not all of which are mediated by Khc (Fig 1D)—and the fraction of Khc-positive mRNPs is proportional to the amount of labelled Khc (Appendix Fig S1G), we assume that the rate of false negative detection is low in this analysis. In contrast to the high degree of association of Khc with motile mRNPs, only ~15% of non-motile *oskar* particles were found to be in complex with Khc during their trajectories in the same analysis. The fact that at any given moment most *oskar* particles are

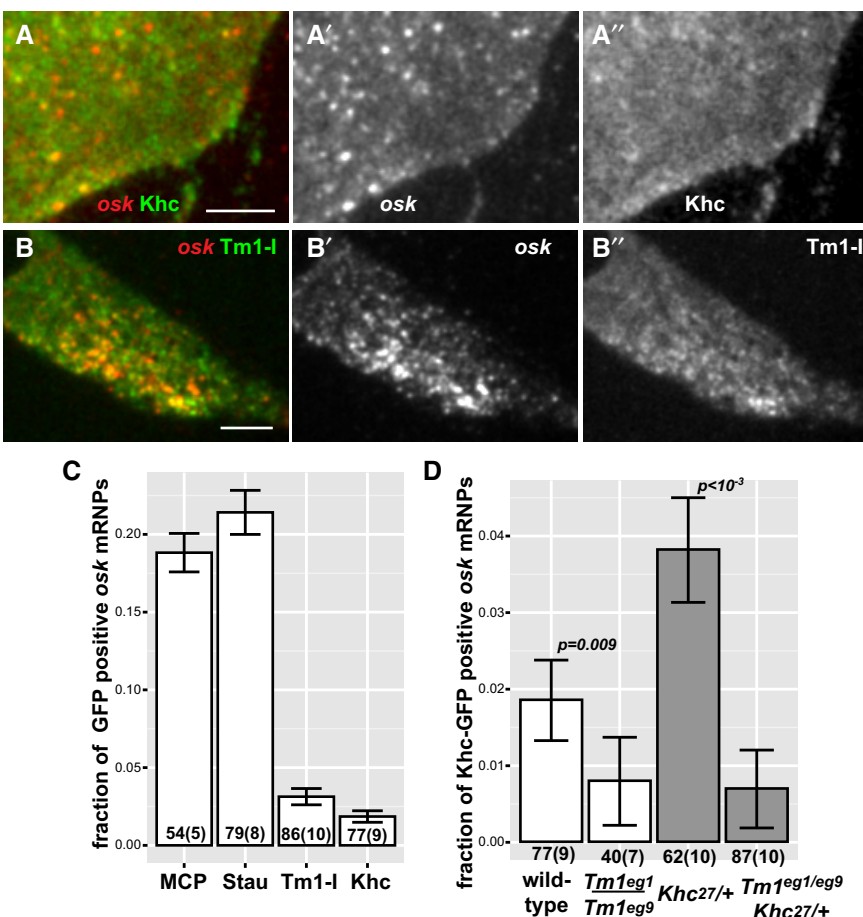

**Figure 2. Composition of *oskar* mRNPs *ex vivo*.**

A–B″  Colocalization of *oskMS2*-mCherry (red, A, B) (A′, B′) with Khc-EGFP (green, A) (A′) or with EmGFP-Tm1-I (green B) (B′) in *ex vivo* ooplasmic preparations. Scale bars represent 5 μm.

C  Fraction of *oskMS2*-MCP-mCherry mRNPs located non-randomly within a 200 nm distance of one of the indicated GFP-tagged protein particles in *ex vivo* ooplasmic preparations. MCP indicates MCP-EGFP which, like MCP-mCherry, can bind to MS2 loops. Staufen (Stau) is a dsRNA binding protein and *bona fide* partner of *oskar* mRNA (St Johnston *et al*, 1991, 1992). All values are significantly different from zero (P < 10⁻³, one-sample *t*-test).

D  Fraction of *oskMS2*-mCherry mRNPs colocalizing (max. 200 nm) non-randomly with Khc-EGFP particles in wild-type and *Tm1$^{eg1}$*/*Tm1$^{eg9}$* ooplasms in the presence of two (white) or one (grey) copy of endogenous Khc.

Data information: (C, D) P-values of two-sample *t*-tests are indicated above the relevant bar pairs. Numbers indicate the number of particle clusters (160 mRNPs in each) and the number of preparations (in brackets) analysed. Error bars represent 95% confidence intervals.

stationary (Video EV1; Appendix Table S1; Zimyanin *et al*, 2008; Ghosh *et al*, 2012; Gaspar *et al*, 2014) indicates that, in accordance with the analysis of snapshot images (Fig 2D), the majority of *oskar* mRNPs are not in complex with Khc in wild-type oocytes.

We also found that ~20% of *oskar* mRNPs are stably associated with mCherry-Tm1-I/C irrespective of their motility (Fig 3B and D). The observed low and nonlinearly scaling proportions of *oskar* mRNPs (Fig 3D) associated with the relatively dim mCherry-Tm1-I (Fig 3B) suggest that (in contrast to Khc-mKate2; see above) the false negative detection rate in this analysis is rather high. Therefore, we cannot reliably determine the true extent of the association and whether one-fifth or a greater proportion of *oskar* mRNPs are associated with Tm1-I/C. Nevertheless, our data show that like Staufen (Fig 3E), Tm1-I/C is a component of *oskar* mRNPs in the oocyte.

Finally, our examination of Khc association with RNPs in mutant extracts lacking Tm1-I/C revealed that it is equally low in the motile and non-motile mRNP populations and that it is considerably below that observed in the wild-type control (Fig 3C). This confirms our analysis of snapshot images and demonstrates that Tm1-I/C is required for proper loading of Khc on *oskar* mRNPs.

## Kinesin-1 associates dynamically with *oskar* RNPs

During stage 9 of oogenesis, half of all *oskar* mRNA molecules in the oocyte translocate to the posterior pole (Gaspar *et al*, 2014). Since only 15% of *oskar* RNPs are in complex with Khc at any given moment (Fig 3C), this implies that kinesin-1 must dynamically redistribute within the RNP population. To assess the possibility of such dynamic recruitment, we measured the frequency of the kinesin-1 and *oskar* association events (Fig 3F). We observed that motile mRNPs associate with a Khc signal about once every 5 s (~0.2/s; Fig 3G) in wild-type ooplasm before and during their motion. This frequency decreased to ~0.1/s in the case of wild-type, non-motile RNPs and dropped to ~0.05/s when Tm1-I/C was absent (Fig 3G). This observation, the low degree of Khc association we observe in $Tm1^{gs}$-mutant ooplasm and the interaction of Tm1-I/C with *oskar* mRNPs (Figs 2C and 3D) indicate that Tm1-I/C acts in recruiting the kinesin motor to the mRNA. The failure of this recruitment explains the greatly reduced number of long, unidirectional—in particular the plus-end-directed—runs of *oskar* mRNPs in the absence of Tm1-I/C (Fig 1D; Appendix Table S1; Zimyanin *et al*, 2008).

## Tm1-I/C is in complex with Khc and *oskar* mRNA and directly binds the *oskar* 3′UTR

If Tm1-I/C is indeed responsible for Khc recruitment to *oskar* mRNPs, these molecules should be in complex with one another. To test this hypothesis, we performed immunoprecipitations from ovarian lysates. Similar to what has been reported previously *in vitro* (Veeranan-Karmegam *et al*, 2016), we detected that Khc specifically co-immunoprecipitated with the EmGFP-Tm1-I bait *in vivo* (Fig 4A). Also, we found Staufen but no other tested *oskar* RNP components (Bruno, Y14, BicD or dynein) in the eluate (Fig 4A). Although this bulk co-immunoprecipitation analysis cannot resolve either the heterogeneity or the spatiotemporal distribution of such complexes, it shows that Tm1-I/C forms complexes with kinesin-1 and with

Staufen that are very likely maintained by—not necessarily direct—protein–protein interactions.

In a screen to identify proteins bound directly to mRNAs in early *Drosophila* embryos, we isolated a non-isoform-specific Tm1 peptide (Sysoev *et al*, 2016). By immunoprecipitating EmGFP-Tm1-I from lysates of embryos exposed to 254-nm UV light, we detected significantly more poly(A)+ RNAs cross-linked to Tm1-I/C than to the control under denaturing conditions (Fig 4B and B′), confirming the RNA binding activity of TM1-I/C. qRT–PCR of cross-linked mRNAs revealed *oskar* as a target of Tm1-I/C (Fig 4C). To identify the region of *oskar* mRNA to which Tm1-I/C binds, we incubated embryonic lysates expressing EmGFP-Tm1-I with exogenous digoxigenin-labelled *oskar* RNA fragments and subjected them to UV cross-linking. Immunoprecipitation allowed the recovery of the 3′UTR, but not other regions of *oskar* mRNA (Fig 4D and E). Truncated (Fig EV3A–C) and non-dimerizing *oskar* 3′UTR (Fig 4F) bound to EmGFP-Tm1-I with greatly reduced affinity. These findings indicate that Tm1-I/C is an RNA binding protein and that its efficient binding to *oskar* mRNPs requires an intact, dimerizing *oskar* 3′UTR.

To test whether Tm1-I/C and Khc co-exist in *oskar* mRNP complexes, we performed *oskar in situ* hybridization on EmGFP-Tm1-I-rescued $Tm1^{gs1}$ egg-chambers carrying one copy of the $Khc^{mKate2}$ allele (Fig 5A–C′). We found that only small portions of *oskar* mRNPs colocalized with Khc-mKate2 (~4.6%) or EmGFP-Tm1-I (~5.7%) in oocytes *in situ* (Fig 4D), similar to what we observed in our *ex vivo* colocalization analysis (Fig 2C). Interestingly, the portion of *oskar* mRNPs positive for both Khc-mKate2 and EmGFP-Tm1-I (~5.1%) was almost 40% higher than the value expected from the amount of colocalization of the mRNA with each component alone ($P = 10^{-4}$; Figs 5F and EV3D). This positive correlation between the presence of Tm1-I/C and Khc in *oskar* transport particles indicates that in most cases when one of the molecules is part of an *oskar* mRNPs, the other molecule is present as well. In a similar analysis, we found that the EJC component Mago, although part of *oskar* mRNPs, did not exhibit such a positive correlation of colocalization with Tm1-I on the mRNA (Fig EV4A and B).

## Kinesin-1 is recruited by Tm1-I/C to *oskar* upon nuclear export

In the course of our FISH colocalization analysis, we noted that Khc-mKate2 and EmGFP-Tm1-I colocalized with *oskar* mRNPs not only in the oocyte, but also in the nurse cell cytoplasm to the same extent (Figs 5A–B′ and G, and EV3E), indicating that the Khc recruiting machinery is already operational in the nurse cells. To test whether this colocalization of Khc with *oskar* mRNA also requires Tm1-I/C, we analysed Khc-mKate2 association with endogenous *oskar* mRNA in $Tm1^{gs}$-mutant nurse cells (Fig 5D and E). We found an almost twofold reduction in Khc-positive *oskar* mRNPs in the absence of Tm1-I/C when a single *Khc* allele was labelled (Fig 5H). When all Khc molecules were fluorescently tagged, we also detected a significant difference in Khc association in $Tm1^{gs}$-mutant and wild-type nurse cells, although we did not observe the almost twofold increase in Khc-positive *oskar* mRNPs in the wild-type controls that we observed in case of the $Tm1^{gs}$ mutant (Fig 5H). This observation highlights the possible limitation of our colocalization analysis when crowding of at least one of the objects occurs (Fig 5D and E).

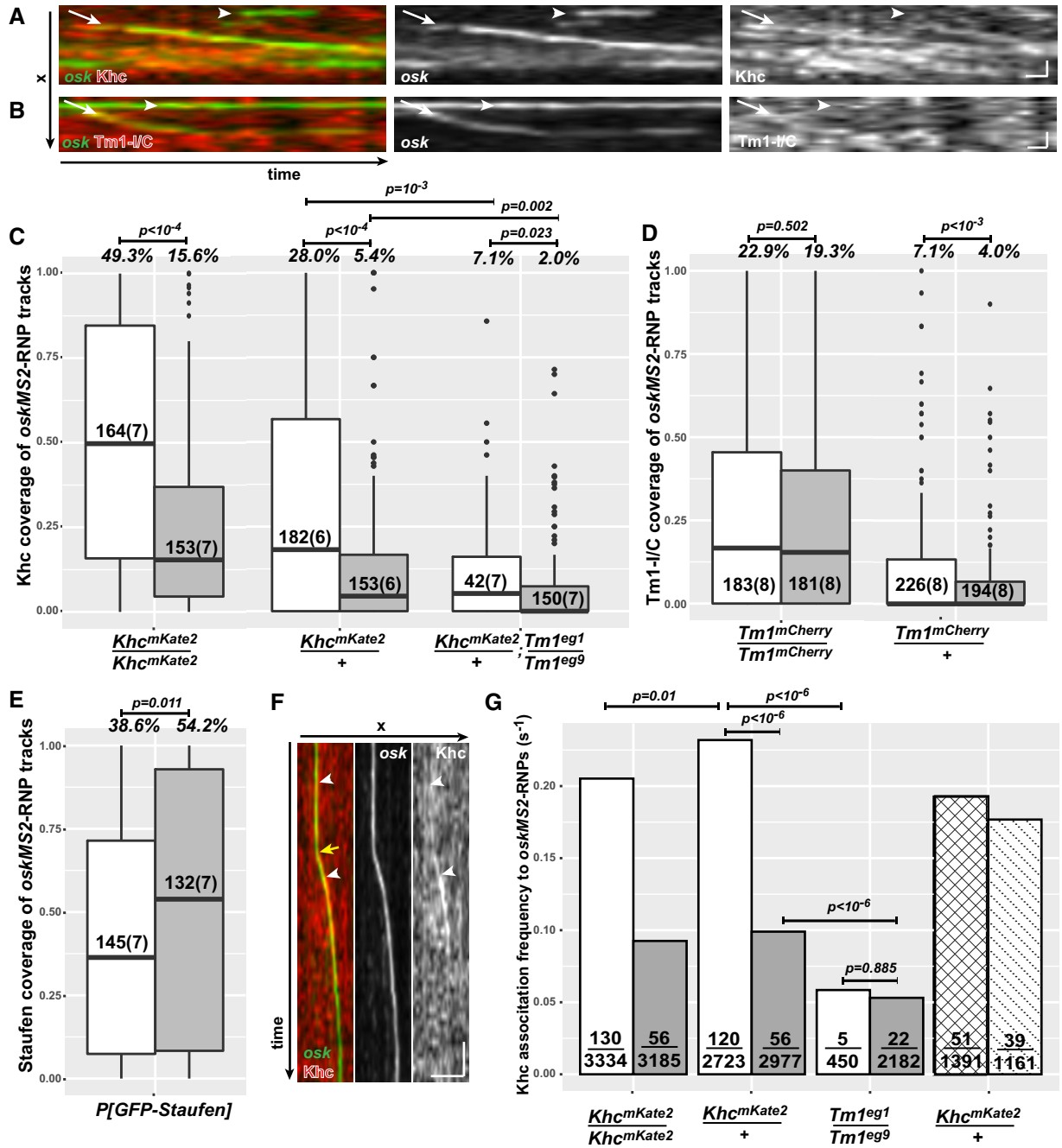

**Figure 3.** **Dynamic composition of *oskar* mRNPs *ex vivo*.**

A, B    Kymographs of *oskMS2*-GFP mRNPs (green) associated with Khc-mKate2 (A, red) and mCherry-Tm1-I/C (B, red) *ex vivo*. Arrows indicate motile RNPs in stable complex with Khc (A) or Tm1-I/C (B), and the arrowheads point to non-motile *oskMS2*-RNPs showing no obvious accumulation of the tagged protein. Note that mCherry-Tm1-I was exposed twice as long as Khc-mKate to obtain comparable red fluorescence signals. Scale bars represent 1 μm and 1 s, respectively.

C–E    Relative Khc-mKate2 (C), mCherry-Tm1-I/C (D) and GFP-Staufen (E) coverage of motile (white) and non-motile (grey) *oskMS2*-GFP trajectories. Numbers within the boxes indicate the number of trajectories and the number of ooplasms (in brackets) analysed. Percentages above the plots show the fraction of RNPs that were found stably and reliably associating with the indicated protein (for at least half of the duration of the trajectory, $P < 0.01$, binominal distribution; see also Fig EV2B). *P*-values of pairwise Mann–Whitney *U*-tests are indicated above the boxplots. The bottom and the top of the box represent the first and third quartiles, the thick horizontal lines indicate the data median. Whiskers show the data range excluding outliers, which are represented by dots.

F    Example kymograph of Khc molecules (red) associating with an *oskMS2*-GFP mRNP (green) before and during its run. White arrowheads indicate two association events, and yellow arrow indicates the onset of motility. Scale bars represent 5 μm and 1 s, respectively.

G    Frequency of Khc-mKate2 appearance on motile (white), before (motility-primed, checked) and after (dotted) the onset of motility, and non-motile (grey) *oskMS2*-GFP trajectories. Fractions within the bars indicate the number of association events that lasted longer than a single frame over the total number of frames analysed. Indicated *P*-values show results of pairwise Fisher's exact test. The Khc association frequency observed on RNPs before (checked) and during (dotted) their motility is not significantly different from wild-type motile RNP controls ($P > 0.01$).

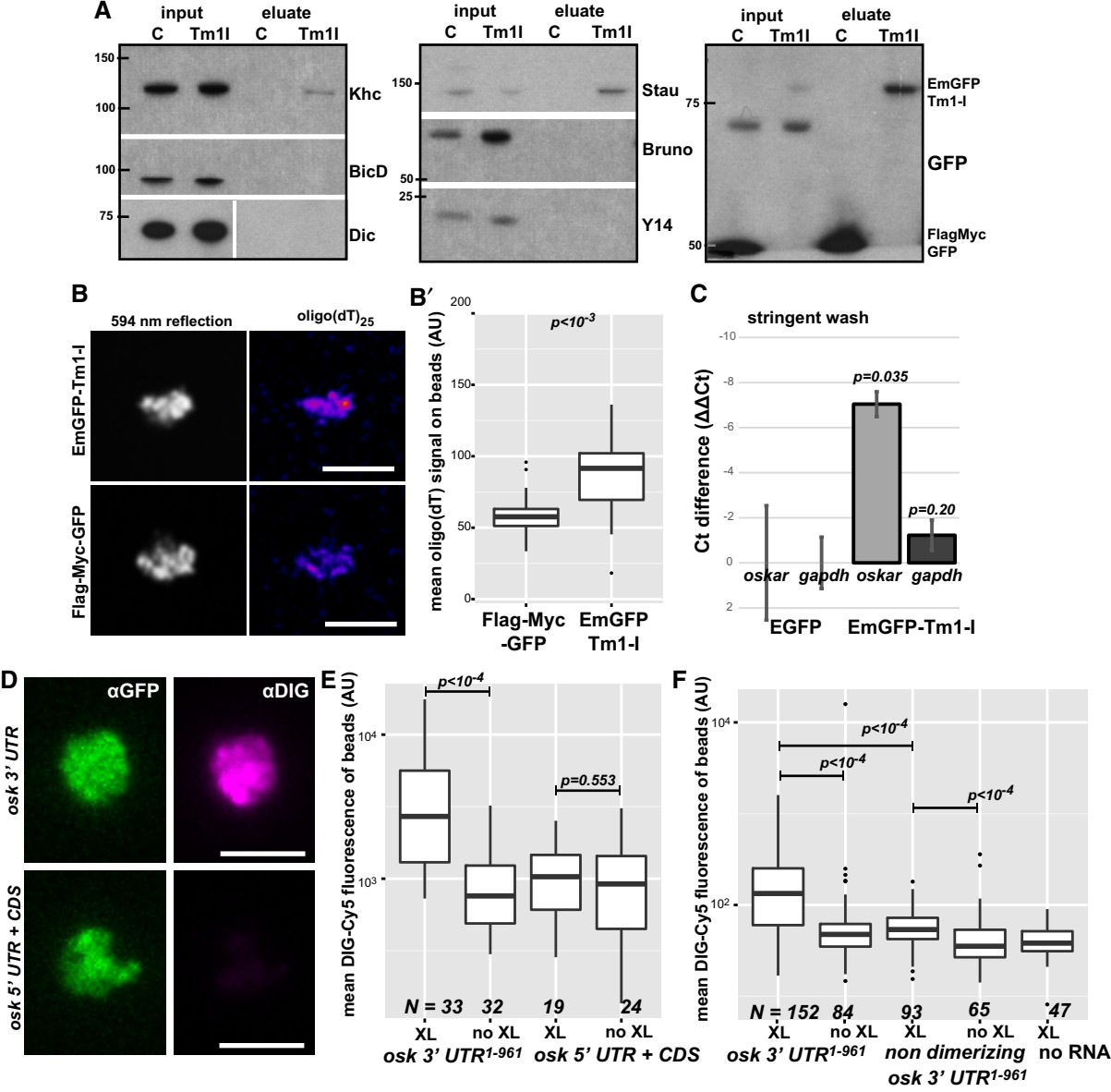

**Figure 4. Tm1-I/C binds RNA and Khc.**

A   Western blots of *oskar* mRNP components (Staufen, Bruno and Y14) and motor-associated proteins (Khc, BicD, Dic) co-immunoprecipitated with EmGFP-Tm1-I from ovarian lysates. Protein marker bands and their molecular weight in kDa are indicated.

B   Signal developed with oligo(dT)$_{25}$–Texas Red probes on GFP-Trap_M beads binding EmGFP-Tm1-I or FlagMycGFP after denaturing washes. *594 nm reflection* marks bead boundaries. Scale bars represent 5 μm.

B'  Quantification of mean oligo(dT) signal measured on beads. *P*-value of a pairwise Mann–Whitney *U*-test is indicated. The bottom and the top of the box represent the first and third quartiles, the thick horizontal lines indicate the data median. Whiskers show the data range excluding outliers, which are represented by dots.

C   qRT–PCR of EmGFP-Tm1-I and control Flag-Myc-GFP and bead alone ($w^{1118}$) bound *oskar* and *gapdh* RNAs after stringent washes (mean ± s.e.m., *N* = 3, two-sample *t*-test).

D   Images of beads binding to EmGFP-Tm1-I (green) and DIG-labelled *in vitro*-transcribed RNA fragments (magenta). Scale bars represent 5 μm.

E, F Mean DIG-Cy5 fluorescence measured on beads capturing the RNA fragment, with or without UV cross-linking, indicated below the charts. *P*-values of pairwise Mann–Whitney *U*-tests are indicated. In panel (F), none of the non-cross-linked samples differ significantly from the no-RNA control (*P* > 0.05). Numbers below the plots indicate the number of beads analysed. The bottom and the top of the box represent the first and third quartiles, the thick horizontal lines indicate the data median. Whiskers show the data range excluding outliers, which are represented by dots.

Source data are available online for this figure.

Although STED super-resolution microscopy further increased crowding by resolving the confocal objects (Fig EV4C, C' and G), it confirmed that both Khc-EGFP and EmGFP-Tm1-I are recruited to *oskar* mRNPs (Fig EV4D). Moreover, it reinforced our finding that Khc-EGFP association with *oskar* mRNPs in the nurse cells is greatly reduced in the absence of Tm1-I/C (Fig EV4E).

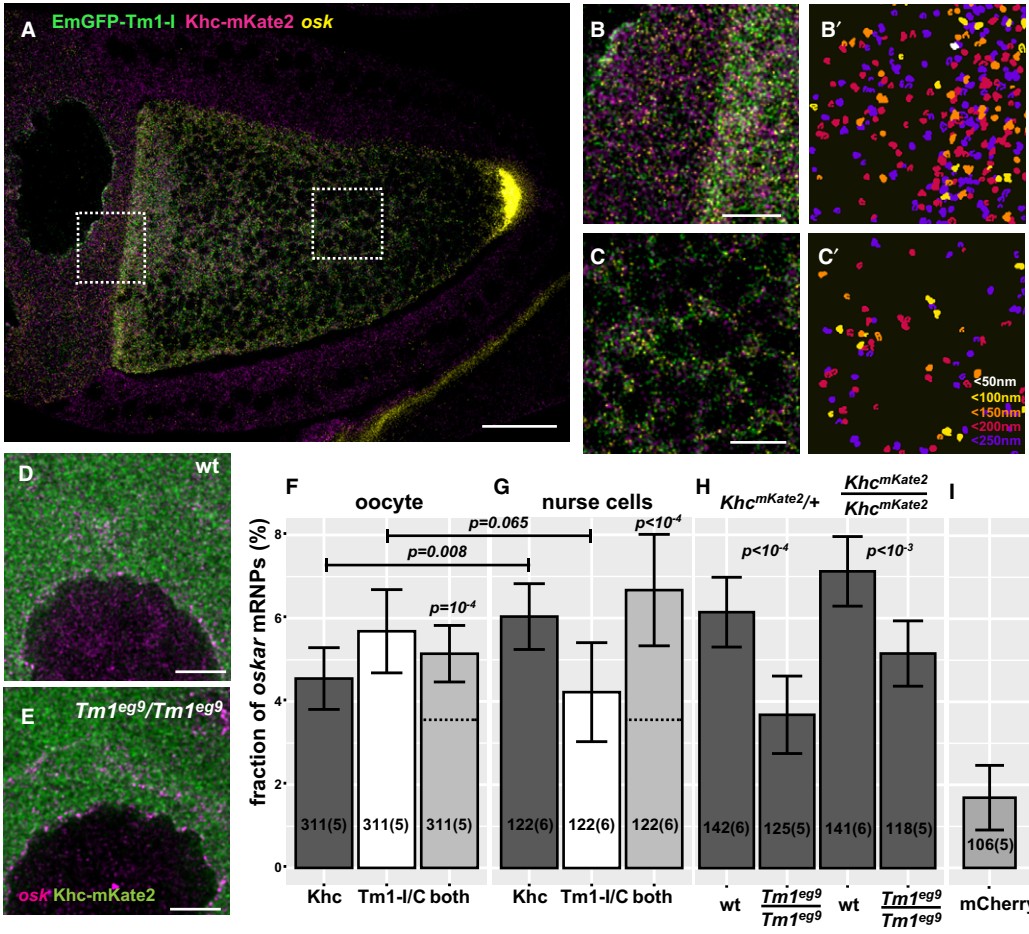

**Figure 5.  Composition of *oskar* mRNPs *in situ*.**

A–C′    Confocal image of a *Tm1[eg9]* homozygous egg-chamber expressing EmGFP-Tm1-I (green) and Khc-mKate2 (magenta). *oskar* mRNA labelled with osk1-5 FIT probes (Hovelmann *et al*, 2014) is in yellow. (B′, C′) *oskar* mRNPs colocalizing with both EmGFP-Tm1-I and Khc-mKate2. Colours indicate the maximal colocalization distance (C′). Panels (B–C′) represent the boxed regions in panel (A).

D, E    Localization of Khc-mKate2 (green) and *oskar* mRNA (magenta) in wild-type and *Tm1[gs]*-mutant nurse cells.

F, G    Fraction of *oskar* mRNPs colocalizing with Khc-mKate (dark grey), EmGFP-Tm1-I (white) or both of these proteins (light grey) in the oocyte (F) or in the nurse cells (G) (max. colocalization distance is 250 nm). None of the values are significantly different from each other (one-way ANOVA, $P > 10^{-3}$). Horizontal dashed lines indicate the expected value of observing both proteins in an *oskar* mRNP if the interactions are independent (see Fig EV3C and D). Significance of the observed colocalization values versus the expected values is shown (one-sample *t*-test). Data obtained from nurse cells and oocytes were compared with pairwise *t*-test.

H    Fraction of *oskar* mRNPs colocalizing with Khc-mKate in wild-type and *Tm1[gs]*-mutant nurse cells when half or all Khc molecules are labelled (as indicated above the graph). *P*-values of pairwise *t*-tests are indicated.

I    Fraction of *oskar* mRNPs colocalizing with free mCherry in wild-type nurse cells used as negative control. The measured fraction (~1.6%) is significantly different from zero (one-sample *t*-test). All other measured colocalization values are significantly different from this negative control ($P < 0.001$, one-way ANOVA).

Data information: (F–I) Numbers indicate the number of particle clusters (100 *oskar* mRNPs in each) and the number of egg-chambers (in brackets) analysed. Error bars represent 95% confidence intervals. All values (F–I) are significantly different from zero ($P < 10^{-3}$, one-sample *t*-test). Scale bars represent 20 μm (A) and 5 μm (B–E).

Fluorescently tagged Tm1-I, similar to Khc-mKate, localized diffusely in the cytoplasm and, unlike other tropomyosins, did not accumulate on actin structures *in vivo* in the egg-chamber (Fig 6C–D′)—although it was shown to bind microfilaments *in vitro* (Kim *et al*, 2011). In contrast, Tm1-I accumulated at the posterior pole of the oocyte (Cho *et al*, 2016; Fig EV5A, C′ and E–G). Interestingly, we also detected the fluorescent Tm1-I signal in the nurse cell nuclei (Figs 6A, EV4F and EV5B); however, in contrast to GFP-Mago, we did not find evidence that nuclear Tm1-I/C associates with *oskar* transcripts (Fig EV4H).

The most prominent localization pattern of the fluorescently tagged Tm1-I was its enrichment around the nuclear envelope (NE) in the nurse cells (Figs 6A and EV5; Cho *et al*, 2016; Veeranan-Karmegam *et al*, 2016), similar to that of *oskar* mRNA (Little *et al*, 2015; Fig 6B). We also detected perinuclear enrichment of endogenous Tm1-I/C on immunolabelled wild-type specimens (Fig EV5G and I), although it was less pronounced due to the high aspecific signal in the nurse cell cytoplasm (Fig EV5H). We observed that Khc also enriches around the nurse cell NE (Fig 6C), although to a lesser extent than the fluorescently tagged Tm1-I/C or *oskar* mRNA

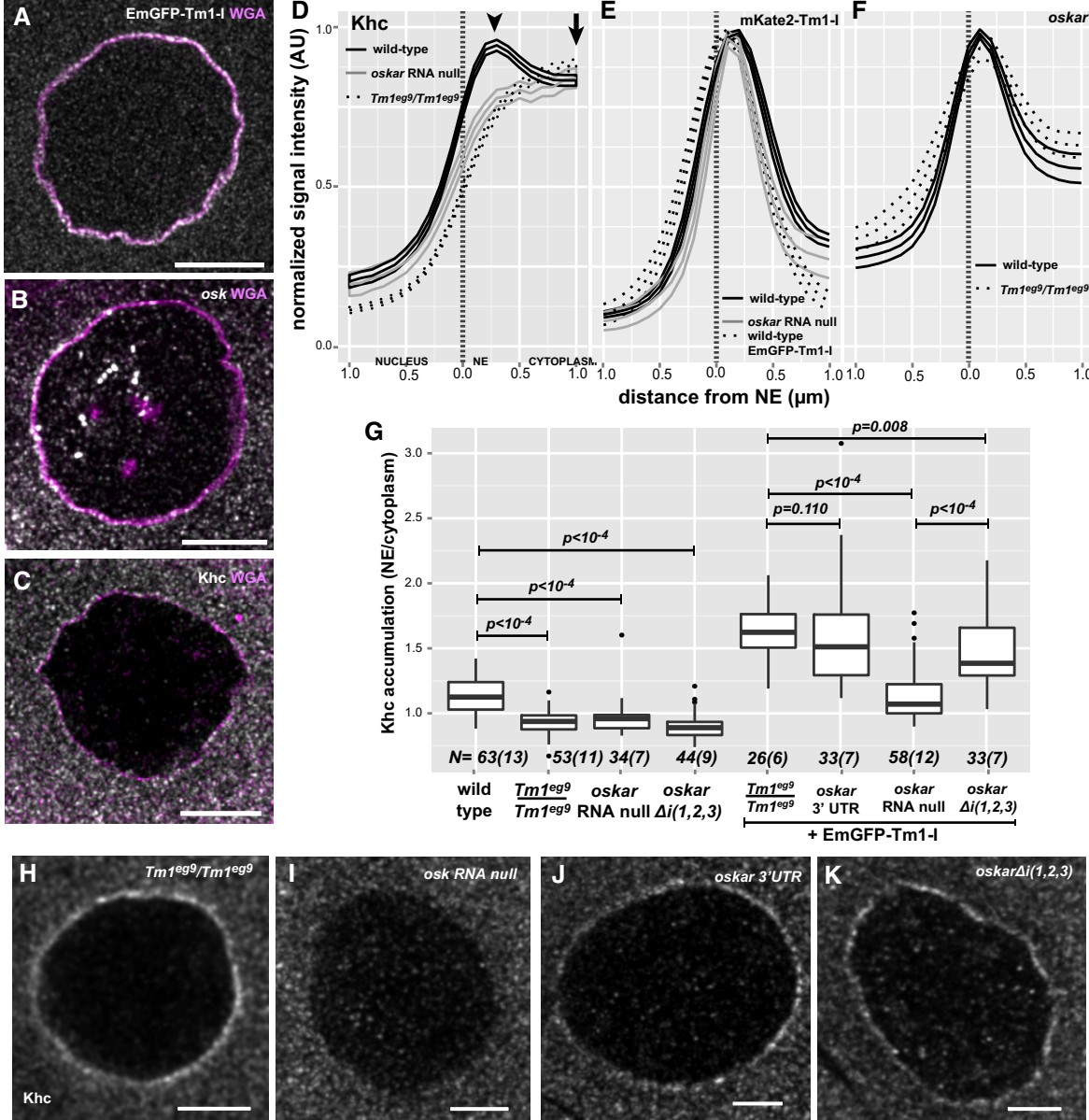

**Figure 6. Accumulation of *oskar* mRNPs around the NE.**

A–C   Localization of EmGFP-Tm1-I (A), *oskar* mRNA (B) and Khc (C) around the nurse cell nuclear envelope (magenta, WGA staining).

D–F   Mean distribution profile of Khc (D) mKate2-Tm1-I, EmGFP-Tm1-I (E) and *oskar* mRNA (F) around the NE of nurse cells (marked with a dotted vertical line). Genotypes are indicated as follows: wild-type control with solid black line (D–F), *oskar* RNA null with solid grey line (D, E) and $Tm1^{eg9}/Tm1^{eg9}$ with dotted black line (D, F). Lines indicate mean and 95% confidence intervals. We analysed the accumulation of mKate2-Tm1-I instead of EmGFP-Tm1-I due to unexpected GFP expression from the $oskar^0$ allele.

G   Khc accumulation around the NE. To calculate accumulation, the signal intensity measured at the position of the peak observed in the wild-type control (D, arrowhead, 356 ± 17.6 nm away from NE) was divided by the signal intensity 2 s.d. away (D, arrow, at 356 + 2*410 nm). *P*-values of pairwise Mann–Whitney *U*-tests against wild-type control or $Tm1^{gs}$ rescued with EmGFP-Tm1-I are indicated above the boxplots. Numbers indicate the number of nuclei and the number of egg-chambers (in brackets) analysed. The bottom and the top of the box represent the first and third quartiles, the thick horizontal lines indicate the data median. Whiskers show the data range excluding outliers, which are represented by dots.

H–K   Khc accumulation around the NE of nurse cells over-expressing EmGFP-Tm1-I within $Tm1^{eg9}/Tm1^{eg9}$ (H), and *oskar* RNA-null egg-chambers (I) expressing either the *oskar* 3′UTR (J) or *oskar* $\Delta i(1,2,3)$ (K).

Data information: Scale bars are 10 μm (A–C) and 5 μm (H–K).

detected by FISH. Our analysis of radial profiles of NEs counterstained by fluorescent lectins confirmed a slight but significant perinuclear accumulation of Khc in wild-type nurse cells (Fig 6D

and G), independent of the developmental age of the egg-chamber (Appendix Fig S2A). This observed Khc accumulation around the NE required the presence of both *oskar* mRNA and Tm1-I/C (Fig 6D

and G). In contrast, *oskar* mRNA or mKate2-Tm1-I accumulation around the NE was not affected in *Tm1^{gs1}* or *oskar*-null mutant egg-chambers, respectively (Fig 6E and F).

We noted that transgenic over-expression of EmGFP-Tm1-I increased Khc recruitment to the NE substantially in the rescued *Tm1^{gs}* egg-chambers (Fig 6G and H). There was a slight elevation in Khc accumulation in the absence of *oskar* RNA (Fig 6G and I), possibly reflecting the ability of the over-expressed EmGFP-Tm1-I to bind Khc on its own, or in the presence of other, even non-specific RNA targets. However, a substantial increase was only observed when an intact *oskar* 3′UTR, whether in endogenous *oskar* mRNA, transgenic non-spliced, non-localizing *oskar* Δ*i(1,2,3)* or the *oskar* 3′UTR alone, was present (Fig 6G, J and K). Given that even in the absence of *oskar* mKate2-Tm1-I was enriched around the NE, these results not only confirm the instrumental role of Tm1-I/C in the Khc recruitment process, but also indicate that kinesin-1 loading on *oskar* mRNPs takes place if and only if mRNAs containing the *oskar* 3′UTR are available. Consistent with this, in the nurse cell cytoplasm of *oskar* RNA-null egg-chambers expressing the *oskar* 3′UTR and Khc-mKate2, we detected an identical degree of Khc association with *oskar* mRNPs as observed in wild-type control egg-chambers (Fig EV4I and J).

### Kinesin recruited by Tm1-I/C is inactive

To address the functional consequences of the "super-loading" of Khc at the NE upon EmGFP-Tm1-I over-expression, we quantified the mean distribution of the intronless, non-localizing *oskar* Δ*i* (*1,2,3*) RNA (Hachet & Ephrussi, 2004; Ghosh *et al*, 2012) through-out stage 9 oocytes (Gaspar *et al*, 2014). This analysis showed that over-expression of EmGFP-Tm1-I causes a substantial posterior-ward shift of the non-spliced *oskar* Δ*i(1,2,3)* mRNA (Fig 7A, B and F). However, the rescue of localization was not complete as it still deviated substantially from the wild-type control (Fig 7C and F). Furthermore, EmGFP-Tm1-I over-expression did not promote posterior localization of an RNA consisting solely of the *oskar* 3′UTR (Fig 7D and F), although the posterior enrichment of Khc was not affected (Appendix Fig S2F and G). EmGFP-Tm1-I over-expression also did not promote *oskar* mRNA localization in oocytes with reduced Khc levels (Fig 7E and F; Appendix Fig S2H), confirming the essential role of kinesin-1 motor is this process. These observations indicate that a properly assembled EJC/SOLE is required to activate the *oskar* 3′UTR-bound, Tm1-I/C-recruited kinesin-1 within the oocyte during mid-oogenesis.

## Discussion

Loading of the appropriate transport machinery on mRNPs is critical to achieve correct localization and, consequently, localized translation of the transcript. Although mRNA transport has been extensively studied over the last two decades, the recruitment of plus-end-directed kinesin motors to mRNPs and their regulation remains poorly understood (Medioni *et al*, 2012).

Here, we have shown that the majority of kinesin-1 motor associated with *oskar* mRNA is recruited by Tropomyosin1-I/C, a non-canonical RNA binding protein, which explains the mislocalization of *oskar* mRNA when Tm1-I/C is lacking (Erdelyi *et al*,

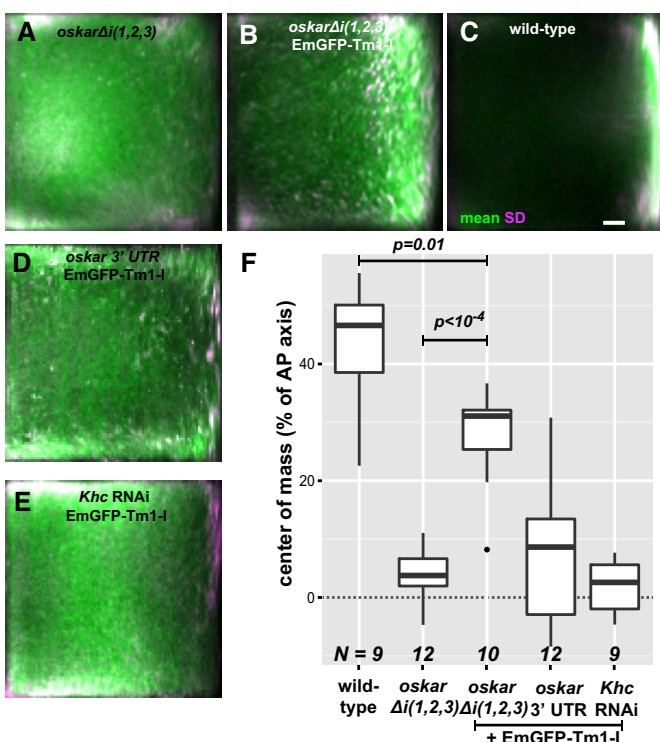

**Figure 7. Effects of EmGFP-Tm1-I over-expression on *oskar* mRNA localization.**

A–E    Mean *oskar* mRNA distribution (green, detected by conventional FISH) within oocytes in which *oskar* mRNA is substituted by *oskar* Δ*i(1,2,3)* mRNA (A, B) and *oskar* 3′UTR mRNA (D) and (in addition) over-express EmGFP-Tm1-I (B, D and E). Wild-type control (C) and oocytes expressing *Khc* RNAi and EmGFP-Tm1-I (E). Magenta indicates standard deviation of the distribution of *oskar* mRNA. Scale bar is 10% of total oocyte length (C).

F    Position of the *oskar* mRNA centre of mass relative to the geometric centre of the oocyte (dotted horizontal line) along the anteroposterior (AP) axis. Posterior pole is the top of the chart. *P*-values of pairwise Mann–Whitney *U*-tests are indicated. Numbers indicate the number of oocytes analysed. The bottom and the top of the box represent the first and third quartiles, the thick horizontal lines indicate the data median. Whiskers show the data range excluding outliers, which are represented by dots.

1995; Veeranan-Karmegam *et al*, 2016; and this study). We found that this recruitment occurs early in the cytoplasmic life of the mRNA, upon its nucleo-cytoplasmic export, in the perinuclear cytoplasm of the nurse cells, where dimerization of *oskar* mRNA molecules via their 3′UTRs commences (Little *et al*, 2015). Our observations are consistent with an "ergonomic" kinesin-loading machinery that becomes functional when and only when Tm1-I/C binds the *oskar* 3′UTR in the nurse cell cytoplasm. The kinesin-recruitment machinery is inefficient, as only a small portion of *oskar* mRNPs are kinesin-bound even during the most active phase of *oskar* mRNA posterior-ward transport in the oocyte. Such inefficiency may serve to prevent the sequestration of kinesin-1 molecules by *oskar* mRNPs away from other cargoes requiring the motor for transport. On the other hand, the transient and dynamic binding and unbinding of the motor may enable transport of virtually all *oskar* mRNPs in a temporally coordinated fashion, thereby promoting localization

of more than 50% of *oskar* mRNA to the posterior pole of the oocyte by the end of stage 9 (Gaspar *et al*, 2014). Additionally, as *oskar* mRNA is continuously transported from the nurse cells to the oocyte from the beginning of oogenesis (Kim-Ha *et al*, 1991; Ephrussi & Lehmann, 1992; Jambor *et al*, 2014), the age of *oskar* mRNPs at the onset of posterior-ward transport may vary between a few minutes and one/one and a half days. The dynamic recruitment of Khc may also guarantee that the *oskar* particles are equipped with transport-competent motor molecules at any moment. It is also apparent from our data that a smaller amount of Khc is recruited to *oskar* mRNPs independent of Tm1-I/C. However, the dynamic exchange of these Khc molecules appears to be rather slow and, most likely as a consequence, they mediate only a minuscule fraction of intra-ooplasmic *oskar* transport—as inferred from the almost complete loss of motility in *Tm1*[gs] mutants (Appendix Table S1) —highlighting the importance of Tm1-I/C in kinesin loading.

The first step of *oskar* transport is mediated by cytoplasmic dynein (Clark *et al*, 2007; Jambor *et al*, 2014), which is presumably also recruited at the nurse cell NE, where we detected the accumulation of the RNA cargo adapter Egalitarian (Appendix Fig S2C; Navarro *et al*, 2004; Dienstbier *et al*, 2009) and the dynactin component dynamitin (Appendix Fig S2D; McGrail *et al*, 1995). Interestingly, the dynein apoenzyme did not enrich around the NE (Appendix Fig S2B and E), possibly because its association with *oskar* mRNPs instantly initiates their transport away from the NE into the oocyte. Within the oocyte, efficient posterior-ward transport of *oskar* mRNA only takes place upon repolarization of the MT cytoskeleton during mid-oogenesis (Theurkauf, 1994; Parton *et al*, 2011). Therefore, the activity of the kinesin-1 recruited to *oskar* mRNPs must be regulated and coordinated with that of the dynein in response either to environmental changes (Gaspar *et al*, 2014; Burn *et al*, 2015) or to the developmental programme. Our data showing the failed or incomplete posterior localization of *oskar* 3′UTR and *oskar* Δi(1,2,3), respectively, indicate that the kinesin-1 recruited to the *oskar* 3′UTR by Tm1-I/C is inactive. Since the *oskar* coding sequence—with the exception of the SOLE—is dispensable for the localization process (Ghosh *et al*, 2012), we propose that during mid-oogenesis, the spliced, EJC-associated SOLE complex activates the *oskar* RNA-bound kinesin-1. Therefore, although the EJC/SOLE is necessary for proper *oskar* mRNA localization, it is not sufficient, as recruitment of the kinesin-1 motor to *oskar* mRNPs is mediated by Tm1-I/C, whose RNA binding scaffold is provided by the *oskar* 3′UTR. To our knowledge, *oskar* is the first mRNA and the first cargo of kinesin-1 described where loading and activation of the motor are decoupled and an unproductive tug of war between two opposing motors is avoided by keeping one of the molecules in a long-term stasis.

Although the underlying mechanisms of the Khc loading and activation processes remain cryptic, a recent study demonstrated that the deletion of the ATP-independent MT-binding site and the auto-inhibitory IAK motif of Khc results in phenotypes consistent with a failure in motor activation and/or recruitment to *oskar* mRNPs (Williams *et al*, 2014). Moreover, it was shown parallel with our study that Tm1-I/C is able to directly bind Khc *in vitro* and that this interaction depends on the ATP-independent MT-binding site (Veeranan-Karmegam *et al*, 2016), indicating that Tm1-I/C directly links kinesin-1 to *oskar* mRNA. This unconventional mode of cargo binding via the unconventional cargo adapter Tm1-I/C may result in incomplete release of kinesin from auto-inhibition and thus explain the persistent inactivity of the kinesin-1 upon its recruitment to the mRNA.

Tm1-I/C is an atypical tropomyosin: it does not enrich around microfilaments in the female germline (Cho *et al*, 2016; Veeranan-Karmegam *et al*, 2016; and this study), forms intermediate filament-like structures (Cho *et al*, 2016) and directly binds to Khc *in vitro* (Veeranan-Karmegam *et al*, 2016) and to RNA, recruiting kinesin-1 dynamically to *oskar* mRNPs. Although Tm1-I/C has a short tropomyosin superfamily domain in its C-terminal moiety, most of the protein is composed of low-complexity sequences (Cho *et al*, 2016). Such intrinsically disordered proteins (IDPs) are often constituents of RNA-containing membraneless organelles, such as RNA granules (Kato *et al*, 2012), stress granules (Molliex *et al*, 2015), P granules (Elbaum-Garfinkle *et al*, 2015) and nuage and germ granules (Nott *et al*, 2015). Furthermore, it has been shown that IDPs fold upon binding to their partners (Wright & Dyson, 2009). This may explain why Tm1-I/C only recruits Khc to the NE in the presence of the *oskar* 3′UTR. Further dissection of the structure and the precise molecular functions of Tm1-I/C and its binding partners will be crucial to determining the nature and the minimal number of features necessary for such unconventional yet vital kinesin-1-mediated localization of mRNA.

# Materials and Methods

### Fly stocks

The *Tm1*[eg1] and *Tm1*[eg9] (FBal0049223) mutants originating from the imprecise excision of the P-element in *Tm1*[gs1] (FBal0049228) were used to study the role of Tm1 in *oskar* mRNA localization (Erdelyi *et al*, 1995). To remove *oskar* mRNA (*oskar* RNA null), a newly created *osk*[attP,3P3GFP] (*oskar*[0] in this manuscript) allele was used in homozygous form or in combination with another RNA-null allele, *osk*[A87] (FBal0141009). We observed no difference between the phenotype of *osk*[attP,3P3GFP] homozygotes and the *osk*[attP,3P3GFP]/*osk*[A87] heterozygotes in our assays. The following *oskar* mRNA mutations and truncations were expressed in an *oskar* RNA-null background: UASp-osk.3′UTR (Filardo & Ephrussi, 2003; FBal0143291) and UASp-oskar Δi(1,2,3) 5x BoxB (Ghosh *et al*, 2012; no protein coding function, FBal0291667). αTub67C::GFP[m6]-Staufen (Schuldt *et al*, 1998; FBal0091177), UASp-dmn-GFP (Januschke *et al*, 2002; FBal0145074), αTub::Khc-EGFP (Sung *et al*, 2008; FBal0230204), UASp-GFP-Mago (Newmark *et al*, 1997; FBal0063884) and Ketel-GFP (Villanyi *et al*, 2008; FBal0244142) were used to visualize Staufen, dynamitin, Khc, Mago and importin-β molecules. To express mCherry FP in the nurse cells, we used the TM3, P{sChFP} balancer chromosome (FBti0141181). The *Khc*[27] protein-null allele (FBal0101625) was used to halve Khc levels (heterozygous) or completely remove Khc from egg-chambers developing from *Khc*[27] homozygous germline clones; the UASp-Khc RNAi Trip Line GL00330 (Staller *et al*, 2013) was used to knock down Khc levels. To label *oskar* mRNPs, we used the *osk::oskMS2*(10x) system together with *hsp83::MCP-EGFP* (Zimyanin *et al*, 2008) or *hsp83::MCP-mCherry* (a gift from L. Gavis) and UASp-EB1-mCherry (a gift from D. Brunner) to label the growing plus ends of MTs. For the

Khc tethering experiment, we expressed *osk::oskMS2(6x)* (Lin *et al*, 2008; FBal0263509), as *oskMS2(10x)* is not translated at the posterior pole (Zimyanin *et al*, 2008). Expression of all UASp transgenic constructs was driven with one copy of *oskar-Gal4* (Telley *et al*, 2012; FBtp0083699), with the exception of the *Khc* RNAi line, where a second *oskar-Gal4* allele was introduced to boost the expression level of co-expressed *UASp-EmGFP-Tm1-I. w^1118^* (FBal0018186) was used as the wild-type control. All stocks were raised on normal cornmeal agar at 25°C. The generation of transgenic lines for this study is described in the Appendix Supplementary Materials.

**Immunological techniques**

Immunoprecipitation of EmGFP-Tm1-I and Flag-Myc-GFP (as control) from ovarian lysates was carried out as described (Ghosh *et al*, 2014). Eluates were tested in Western blot analysis probing with anti-Khc (1:5,000; Cytoskeleton), anti-Staufen (1:1,000) (Krauss *et al*, 2009), anti-Bruno (1:5,000) (Chekulaeva *et al*, 2006), anti-Y14 (1:2,500) (Hachet & Ephrussi, 2001), anti-BicD (2:100, DHSB clone #1B11 and 4C2), anti-Dic (1:2,000; Millipore) and anti-GFP (1:2,000; Millipore). The same antibodies and dilutions were used to detect RNAi knockdown efficiency and relative levels of Khc-EGFP. To test Tm1-I/C expression in ovarian lysates, a pan-Tm1 antibody (1:1,000) (Cho *et al*, 2016) recognizing the tropomyosin domain shared by the Tm1 isoforms was used (a gift from D. Montell).

Khc, Tm1, Dic and Egl were visualized in heat-fixed egg-chambers (Gaspar *et al*, 2014) incubated overnight at 4°C in anti-Khc (1:250), pan-Tm1 (1:500) (Cho *et al*, 2016), anti-Dic (1:1,000) or anti-Egl (1:1,000) (Navarro *et al*, 2004) primary antibodies diluted in PBST (PBS + 0.1% Triton X-100)/10% normal goat serum. Signal was developed by applying AlexaFluor 488- or Cy5-conjugated anti-rabbit or anti-mouse secondary antibodies for 60 min at room temperature (RT) (1:1,000; Jackson ImmunoResearch). GFP autofluorescence was preserved by fixation for 20 min in 2% PFA/0.05% Triton X-100 in PBS. To visualize F-actin, samples were fixed in 2% PFA/0.05% Triton X-100 in BRB80 (80 mM PIPES, 2 mM MgCl$_2$, 1 mM EGTA) and stained with phalloidin–FITC (1:100; Molecular Probes), and all washes were carried out in BRB80 + 0.1% Triton X-100.

NE was counterstained by WGA-TRITC (1:300; Life Sciences) for 60 min at RT. Samples were embedded in 80% glycerol + 2% N-propyl-gallate mounting medium.

**RNA recovery after cross-linking**

To test whether Tm1-I/C binds RNA directly, 0 to 2-h-old embryos expressing EmGFP-Tm1-I, EGFP or FlagMycGFP were collected and 1 J/cm$^2$ UV (λ = 254 nm) was applied to cross-link nucleic acids with protein molecules at zero distance. Embryos were lysed in 10× RIPA buffer composed of low-salt buffer (20 mM Tris–Cl, 150 mM KCl, 0.5 mM EDTA, pH = 7.5) containing strong detergents (1% Triton X-100, 1% deoxycholate and 0.1% SDS), reducing agent (5 mM DTT) and 5× protease inhibitor cocktail (PIC; Roche), and crude debris was removed by 10-min-long centrifugation with 16,000 *g* at 4°C. Clarified lysates were diluted to 10 mg/ml with 10× RIPA buffer and then further diluted tenfold with low-salt buffer. 1 mg protein of lysate (1 ml) was supplemented with 10 µl of GFP-Trap_M beads (Chromotek), and immunoprecipitation

was carried out at 4°C for 90 min. Beads were washed six times with high-salt buffer (20 mM Tris–Cl, 500 mM NaCl, 1 mM EDTA, 0.1% SDS, 0.5 mM DTT, 1× PIC pH = 7.5) at room temperature for a total duration of 60 min (stringent washes). Alternatively, beads were washed twice with high-salt buffer and then three times under denaturing conditions with 7.6 M urea and 1% SDS in PBS (Chromotek Application Note). RNA was then extracted from the washed beads, and qRT–PCR or FISH was performed on the beads with oligo(dT)25–Texas Red (Castello *et al*, 2012). FISH-on-beads experiments were imaged on a Leica SP8 with a 63× 1.4 NA oil-immersion objective. Bead contours were captured by imaging the reflection of 594-nm light, and oligo(dT)25 probes were excited with 594 nm. Primer sequences used during qRT–PCR are available upon request.

To analyse *oskar* mRNA fragments binding to EmGFP-Tm1-I, embryos were lysed in pre-XL buffer (20 mM Tris–Cl, 150 mM KCl, 4 mM MgCl$_2$, 2× PIC, pH = 7.5) and cleared from debris, and lysate concentration was adjusted to 10 mg/ml. 0.2–0.8 ng/nucleotide DIG-labelled RNA (1:10 ratio of labelled to unlabelled uracil) and 1 µl RiboLock (ThermoFisher Scientific) were added to 100 µl lysate. Mixtures were incubated for 20 min at RT prior to cross-linking with 254-nm UV light, 0.15 J/cm$^2$ energy. Subsequently, the lysate was diluted with an equal volume of low-salt buffer supplemented with detergents, reducing agent and PIC (see above) and further diluted 1:5 with low-salt buffer. 2 µl RiboLock and 5 µl GFP-Trap_M beads were added per 500 µl volume, and immunoprecipitation and washes were carried out as described above. Beads were then labelled with anti-GFP-CF488A (1:1,000; Sigma-Aldrich) and anti-DIG-HRP (1:500; Roche) PBST for 30 min. Anti-DIG signal was developed using Cy5-TSA amplification kit (PerkinElmer). Fluorescently labelled beads were mounted on slides in glycerol-based mounting medium and were imaged with a Leica 7000 TIRF microscope using a 100× 1.46 NA oil-immersion objective, 1.6× Optovar and epifluorescent illumination.

**Fluorescent *in situ* hybridization**

For conventional FISH, specimens were fixed in 2% PFA/0.05% Triton X-100 in PBS for 2 h at RT. After fixation and washes in PBST, 2 µg/ml proteinase K was applied for 5 min at RT, followed by 5 min boiling at 92°C in PBS/0.05% SDS. The samples were then pre-hybridized for 60 min at 65°C in hybridization buffer (HYBEC: 5× SSC, 15% ethylene carbonate, 50 µg/ml heparin, 0.1 mg/ml salmon sperm DNA, 0.05% SDS, pH = 7.0). The *oskar* cds (targeting nucleotides 1,442–1,841) and *oskar* 3′UTR (targeting 2,342–2,730) probes were diluted in HYBEC in 0.5 pg/nucleotide/ml concentration each. The *oskar* cds probes were directly labelled with either Atto-488 (Atto-tec) or AlexaFluor 555 (ThermoFisher Scientific), while the *oskar* 3′UTR was directly labelled with Atto-633 (Atto-tec). Hybridization was carried out at 65°C overnight and excess probe was removed by four washes at 65°C (HYBEC, HYBEC/PBST 1:1, 2× PBST, each 20 min) and one 20-min-long wash in PBST at RT. During conventional FISH, WGA-FITC was applied to counterstain NEs. Samples were embedded in 80% glycerol + 2% N-propyl-gallate mounting medium.

To preserve GFP and mKate2 autofluorescence, forced intercalation (FIT) probe-based RNA detection was performed (Hovelmann *et al*, 2014). Ovaries were fixed for 20 min in 2% PFA/0.05% Triton

X-100 in PBS, and they were washed twice for 10 min in IBEX (10 mM HEPES, 125 mM KCl, 1 mM EDTA, 0.3% Triton X-100, pH = 7.7) after fixation. *oskar* mRNA was labelled with osk1-5 LNA-modified FIT probes (Hovelmann *et al*, 2014) in IBEX+ (IBEX supplemented with 15% ethylene carbonate, 50 µg/ml heparin and 10% 10-kDa dextran sulphate) to a final concentration of 0.05 µM each. Samples were incubated at 42°C for 30 min and then briefly washed in IBEX and IBEX/BRB80 (1:1 mixture) at 42°C.

To detect *oskar* 3′UTR with single-molecule FISH (smFISH), we used 15 different 3′UTR-targeting probes labelled with a single Atto-565-ddUTP nucleotide using TdT (Appendix Table S2). smFISH was carried out similar to conventional FISH. Proteinase K digestion and heat denaturation of RNA secondary structures were omitted to preserve mKate2 fluorescence. The hybridization was performed at 37°C for 2 h using 5 nM/probe concentration.

Specimens were embedded in 79% TDE (η = 1.475; Staudt *et al*, 2007), which boosted the brightness of GFP and mKate2 by about twofold and of FIT probes four- to six-fold compared to conventional glycerol-based mounting media (data not shown).

## Microscopy

Conventional laser scanning confocal microscopy was carried out using a Leica TCS SP8 microscope with a 63× 1.4 NA oil-immersion objective. STED microscopy was performed on a Leica STED 3× microscope with a 100× 1.4 NA oil-immersion objective and HyD time-gated photodetectors. Acquired images were deconvolved with Huygens Professional (SVI) prior to analysis.

To minimize crosstalk of the two labels, GFP- and TO-labelled FIT probes were excited with 470-nm and 525-nm lines of a white-light laser source, respectively. Emission was recorded between 480 and 520 nm (GFP) and 525 and 575 nm (TO). For similar reasons, Atto-565 and mKate2 were excited by 561-nm and 610-nm light and emitted fluorescence was detected between 565 and 585 nm and between 620 and 720 nm, respectively. Under these conditions, < 1% of recorded signal originated from crosstalk.

To stimulate emission of the GFP and TO dyes, a 592-nm depletion doughnut-shaped laser beam was used, with all power assigned to improve lateral resolution by about 2.5- to 3-fold. Typically, a stack of seven slices was recorded (voxel size: 22 × 22 × 180 nm) and subsequently deconvolved with Huygens Professional. The middle slice was then subjected to object-based colocalization analysis.

## Ex vivo ooplasmic preparation

Crude ooplasm was obtained from living stage 9 oocytes expressing *oskMS2(10x)*, MCP-EGFP, oskGal4 and EB1-mCherry for mRNP tracking or *oskMS2(10x)*, MCP-mCherry and a GFP-tagged protein of interest for *ex vivo* colocalization analysis. Ovaries were dissected in BRB80 (80 mM PIPES, pH = 6.9, 2 mM MgCl₂, 1 mM EGTA). BRB80 was replaced with 1% IB (10 mM HEPES, pH = 7.7, 100 mM KCl, 1 mM MgCl₂, 1% 10-kDa dextran), and ovaries were transferred onto silanized coverslips. Silanization was carried out with dichlorodimethylsilane (Sigma-Aldrich) under vacuum for 1.5–2 h to obtain a slightly hydrophobic surface. A drop of Voltalef 10S oil (VWR) was placed next to the dissected ovaries, and individual ovarioles containing stage 9 egg-chambers were pulled under oil with fine tungsten needles. There, the stage 9 egg-chambers were

isolated and the nurse cell compartment was carefully removed with needles. Using a gentle pulling force at the posterior pole of the created "oocyte sack" (the oocyte and surrounding follicle cells), the ooplasm was slowly released from anterior to posterior onto the coverslip surface. The level of surface hydrophobicity was critical: a hydrophilic surface bound *oskar* mRNPs aspecifically, blocking their motility, whereas it was impossible to create an ooplasmic streak on coverslips that were too hydrophobic. Such preparations were imaged on a Leica 7000 TIRF microscope with a 100× 1.46 NA oil objective and 1.6× Optovar. Images were collected simultaneously for 32 s with a Photometrics Evolve 512 EM CCD camera with 140 nm lateral resolution. We observed no decline in RNP motility and MT dynamics within the first 60 min (data not shown).

## Image analysis

*In vivo* tracking of *oskMS2*-GFP particles and *oskar* mRNA distribution within oocytes was performed as previously described (Gaspar *et al*, 2014). Image segmentation for tracking, colocalization analysis of *oskar* mRNPs and measurement of bead fluorescence was carried out using a custom particle detector library in ImageJ.

### Extraction of NE radial profiles
A section containing a close-to-maximal cross section of the nucleus was selected, and the outline of the NE counterstained with WGA was coarsely traced manually. At each point along the outline, the signal under a 5-µm-long segment perpendicular to the outline was extracted, resulting in a few hundred to thousand, roughly 2.5-µm-long reads on both the cytoplasmic and the nuclear side of the outline. At each point, the position of the NE was determined with sub-pixel precision through fitting a Gaussian function to the WGA signal. All other recorded signal was positioned relative to the location of the NE and was averaged for a given nucleus. These mean signal intensities were then normalized to the maximum value of a radial profile.

### Ex vivo tracking
*Ex vivo* tracking was done automatically. Tracks displaying linear displacements were manually selected and their directionally manually assigned by overlaying them with the EB1 channel. 8–20% of detected tracks could not be assigned a polarity due to the absence of nearby co-axial EB1 comets (Fig EV1B). Linear runs were extracted from the detected tracks using a series of custom Excel macros (Gaspar *et al*, 2014). Runs of unknown polarity were on average shorter than classified minus- and plus-end-directed runs (Fig EV1D).

### Object-based colocalization
Object-based colocalization was assayed by measuring the distance between closest-neighbour objects from the *oskar* mRNP (reference) and GFP/mKate2 (target) channels within a confined area representing exclusively the nurse cell perinuclear region and cytoplasm. Random colocalization was addressed by seeding the objects of the target channel randomly into the confined area. This simulation was repeated one hundred times to obtain a distribution of expected (random) colocalization. To calculate fraction of colocalization and to compensate for the huge variability of observed particles per image, reference channel objects were randomly assigned into particle clusters representing 160

particles for *ex vivo* and *in situ* colocalization analysis, and 100 particles for competitive FISH. With these values, only ~5% of observed objects of the reference channel were excluded from the analysis. Observed colocalization within each cluster was compared to the distribution of simulated random values using one-sample Student's *t*-test ($\alpha = 0.01$). Significant values were used to calculate the difference between observed and random colocalization to assess true, biological colocalization. These differences were found to be significantly different from zero for all analysed protein molecules in wild-type samples, except for the negative control Ketel-GFP (Fig EV4D). By opening the colocalization window (the maximal inter-neighbour distance), random colocalization rapidly overcomes the observed values (e.g. Fig EV1E–H) due to particle crowdedness both *ex vivo* and *in situ* resulting in a probable underestimation of true colocalization. To minimize this effect, the clipping point where the difference was maximal was determined, and the halfway distance between zero and the clipping point was used to compare the effects of different conditions on *oskar* mRNP composition. This colocalization window was 200 nm *ex vivo* (non-fixed specimen), 250 nm *in situ* and 100 nm STED *in situ* (fixed).

*Temporal colocalization*

Temporal colocalization was assayed as described in the legend to Fig EV2A. Importantly, the same microscope settings were used to acquire image sequences of a given protein molecule (e.g. Khc-mKate2) that allows direct comparison of signal intensities between hetero- and homozygous extracts (see thresholding in the legend to Fig EV2A).

## Statistical analyses

Transformations and statistical analysis of all the obtained numerical data were carried out in R (Team, 2012) using the R Studio (https://www.rstudio.com/) front-end and ggplot2 library (Wickham, 2009) to plot the graphs. Normal distribution of the sampled values was determined by Shapiro–Wilk test. Alpha values for statistical tests were chosen based on average sample size as follows: $\alpha = 0.05$, $1 < N \leq 10$; $\alpha = 0.01$, $10 < N \leq 100$; and $\alpha = 0.001$, $100 < N$. Sample sizes (pooled from at least two replicates) are indicated in the figures and figure legends. Only two-sided statistical tests were used. Measurements that were not within the [Q1 (first quartile) − 1.5 × (Q3 (third quartile) −Q1), Q3 + 1.5 × (Q3−Q1)] range of the samples were scored as outliers.

**Expanded View** for this article is available online.

## Acknowledgements

We thank Denise Montell for sharing unpublished data, antibodies and discussions. We thank Damian Brunner, Tze-Bin Chou, Elizabeth Gavis, Antoine Guichet, Daniel St Johnston and the Developmental Studies Hybridoma Bank for fly stocks and reagents, and the TRiP at Harvard Medical School (NIH/NIGMS R01-GM084947) for transgenic RNAi fly stocks used in this study. Stocks obtained from the Bloomington *Drosophila* Stock Center (NIH P40OD018537) were used in this study. We are grateful to Frank Wippich and Simon Bullock for their comments on the manuscript. We thank Sandra Müller and Alessandra Reversi for fly transgenesis, the EMBL Advanced Light Microscopy Facility and Leica for providing cutting-edge microscopy and the EMBL Genomics Core Facility for their help with qRT–PCR. This work was funded by the EMBL.

## Author contributions

IG and AE conceived the experiments and wrote the manuscript. VS carried out qRT–PCR analysis of mRNAs immunoprecipitated under stringent conditions. AK carried out the EmGFP-Tm1-I co-immunoprecipitation assays. IG carried out the rest of the experiments and data analysis.

## Conflict of interest

The authors declare that they have no conflict of interest.

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
