## [Review Process File · The EMBO Journal]

Manuscript EMBO-2016-96038

An RNA-binding atypical tropomyosin recruits kinesin-1 dynamically to *oskar* mRNPs

Imre Gaspar, Vasilij Sysoev, Artem Komissarov, and Anne Ephrussi

Corresponding authors: Imre Gaspar and Anne Ephrussi, EMBL

Review timeline:

Submission date:	07 November 2016
Editorial Decision:	23 November 2016
Revision received:	28 November 2016
Accepted:	29 November 2016

Editor: Ieva Gailite

Transaction Report:

(Please note that the manuscript was previously reviewed at another journal and the reports were taken into account in the decision making process at The EMBO Journal. Since the original reviews are not subject to EMBO's transparent review process policy, the reports and author response cannot be published.)

1st Editorial Decision

23 November 2016

I have now received two referee reports on your manuscript that are shown below. As you can see from the comments, both referees appreciate your study and support publication at The EMBO Journal, pending satisfactory minor revision.

The referees raise a number of concerns that have to be addressed to allow publication here. Importantly:

1. The referees state that the data presentation is currently confusing and we agree with this point. Therefore, the suggestions of the reviewers regarding the presentation of data and textual clarifications need to be addressed in full.
2. Referee 1 requests additional data on association between Khc and *oskar* mRNP (points 1-2). Please include the data if available, or adjust the data interpretation and discussion accordingly.
3. Referee 1, point 6 - please include the version of Figure 7 from the Biorxiv version in the manuscript for clearer data presentation.
4. Please cite and discuss the similarities and differences of the presented model to the one offered in the manuscript of Gonsalvez and colleagues.

I would therefore like to invite you to submit a revised version, addressing all concerns raised by the referees and especially the ones mentioned above. Should you be able to do so, I can offer to accept the manuscript for publication at The EMBO Journal.

Keeping in mind the situation, we would like to have the manuscript online before the end of the year. We therefore will need to transfer the manuscript to our publisher by December 1st, and we thus would need to receive the modified version of the manuscript by this date. I realise that this gives you an extremely limited amount of time to address the revision points, so please notify me

whether you find it possible to complete the revision until this deadline. Also please contact me if you would like to discuss any other aspect of the revision.

Thank you again for giving us the chance to consider your manuscript for The EMBO Journal, I look forward to receiving your revision in the near future.

REFEREE REPORTS

Referee #1:

Kinesin motors play an important role in transporting mRNAs within the cytoplasm in many cell types. How mRNAs are linked to kinesins is, however, poorly understood. The manuscript provides evidence that a product of the tropomyosin locus (TM1-I/C) provides a long sought after link between oskar mRNA and the kinesin-1 motor during *Drosophila* oogenesis. The importance of this work extends beyond the mRNA localization field to the motor field; data is presented that supports a novel mechanism in which kinesin-1 is bound to cargo in an inactive form to be activated at a specific stage. The authors propose that this is a mechanism that minimizes interference with dynein motors that transport oskar mRNA from the nurse cells to the oocyte (until mid-oogenesis when kinesin-1-based transport is activated).

Several of the findings (e.g. on RNA binding activity and motor inhibition/activation) go beyond what was addressed in the recently published paper by Gonsalvez and colleagues that also proposed that the TM1 isoform is a link between oskar mRNA and kinesin-1 (note that my review takes into consideration that the Gaspar paper has been granted scoop protection by EMBO Journal). The data examining colocalization of TM1-I/C, Khc and oskar mRNA in timelapse movies of *ex vivo* transport assays, which appear to have been produced in response to the comments of the previous reviewers, make a more convincing case for some of the proposed functions of TM1-I/C than the data in Figure 2. However, the manuscript is still difficult to follow in places and there are still some significant issues that should be addressed by changes to the text, figures and movies.

Major points:

1. The authors do a commendable job of controlling for false positives in their analysis of colocalization data. I am less convinced that they are able to control for false negatives, i.e. complex formation being underestimated due to high levels of free proteins in the cytoplasm and/or some complexes containing a relatively low copy number of the fluorescently labeled protein. The data in Figure 3A and B, for example, appear to illustrate the difficulties in dismissing binding of one component to a motile complex containing oskar mRNA (due to high cytoplasmic protein signal). This is significant as some of the conclusions are based on the absolute colocalization frequencies (the conclusions examining relative differences in different genotypes are, of course, not affected).

To give an example, the authors use the value of 15% of oskar mRNPs having detectable enrichment of Khc to argue for a model in which kinesin-1 dynamically turns over on the mRNP population. Do the authors have data that provides a convincing case that Khc is not bound to oskar mRNPs much more frequently (for example after an initial recruitment process at the nuclear envelope), and that colocalization can only be detected above background in 15% of cases? I think that this alternative model should be discussed or data presented/discussed that argues against it. At the moment the authors state in the abstract that a small but dynamically changing subset of oskar mRNPs gets loaded with inactive kinesin-1 and this statement is not fully justified by the current manuscript in my opinion (see next point).

2. The data that (the authors claim) backs up the dynamic association model by purporting to show association of Khc with mRNPs that precedes transport (Figure S3B) is weak. To back up this claim it would be necessary to show data from timepoints even earlier than those shown in Figure S3B, when one might expect Khc levels to be lower than on the transported mRNP. Even if that is the case, it is not clear that the levels of enrichment are significant, especially as a similar level of enrichment is seen for TM1-I/C. By the authors' own admission, the number of events that can be scored limits the power of this analysis. Unless this analysis can be strengthened by more data it should be omitted from the manuscript. This section of the manuscript would also be stronger if several example images of dynamic association of Khc with oskar mRNPs (quantified in Figure 3F)

were provided in the supplementary section.

3. Why are mRNPs that colocalize with TM1-I/C no more likely to exhibit motile behaviour than non-motile behaviour (Figure 3D)? The authors should consider offering an explanation/hypothesis for this observation. Could it be that TM1-I/C is a constitutive component of the mRNP (necessary for Khc binding) but that scoring colocalization is difficult because of background levels of the protein in the cytoplasm?

4. I agree with Reviewer 1 that calling TM1-I/C a tropomyosin or an atypical tropomyosin could cause some readers confusion. Does this protein encode a coiled coil, which is a key feature of tropomyosins? Even if it does, it may be best to be conservative and refer to the protein as a product of the tropomyosin locus in the title and abstract, as many readers won't look into the manuscript for details.

5. Figure S4A. Were these data derived from the same experiments presented in the text with the longer 3'UTR fragment? If so, the data for the longer 3'UTR should be shown for comparison in Figure S4A. If not, it seems that concluding that the 3'UTR truncations compromise RNA binding is not justified.

6. The Biorxiv version of the paper includes an additional way of presenting the results in Figure 7 of the current manuscript (Figure 7L in the preprint), which is very helpful for understanding the dataset. I recommend that this is reinserted in the manuscript. The preprint also has a clearer description of the advantage of tagging endogenous proteins when evaluating colocalization.

Minor points:

1. I also agree with the reviewer's comment that some information on the features of the TM1-I/C protein (relationship to tropomyosin/presence of low complexity regions) should be introduced in the Results so that the reader can keep this in mind when reading the rest of the manuscript. I suggest that the authors consider this issue again.

2. It wasn't clear to me if the authors currently think that TM1-I/C is the only recruiter of Khc, if there could be an independent mechanism for recruiting some Khc, or if this question needs more work to resolve in the future (e.g. there is a subheading at the beginning of the Results stating that TM1-I/C 'maintains' kinesin-1 on oskar mRNAs).

3. TM1-I/C 'acts upstream' of Khc. This genetic term should be avoided when talking about a biochemical complex.

4. Figure 2A and B. The data would presumably be clearer if the single color images are shown in this figure or a supplementary figure.

5. "Although such an en mass co-IP analysis lacks a spatiotemporal resolution, it shows that Tm1-I/C, kinesin-1 and Staufen indeed form complexes...". The analysis shows that Tm1-I/C forms complexes with kinesin-1 and that it forms complexes with Staufen. Although it is likely to be the case, the data do not show that all three components are in the same complex. The authors should consider modifying their language to make this clearer.

6. Figure 6 title. 'Transient' is not necessary as the figure does not address the persistence of the oskar association at this site.

7. The nature of oskar delta I (1,2,3), and its relationship to the SOLE/EJC, needs to be explained when first introduced. This is essential for understanding the logic of this very informative experiment.

8. Figure S7C and D should be called in the text when referring to the enrichment of Egl and p50 around the nuclear envelope.

9. References need to be provided in the methods for the pan-TM1 and Egl antibodies.

10. Figure S4B. What is FMG?

11. Figure S4C and D legend. The text does not match the panels (rather it describes data in Figure S5). I therefore couldn't deduce what S4C and D show.

12. I don't understand what Figure S5C and C' are supposed to show as not much information is given about this experiment in the text. Also, in the corresponding legend, 'auto-fluorescence' is referred to for GFP. Presumably 'fluorescence' could be used to prevent potential confusion with cytoplasmic autofluorescence?

13. Is Figure S5D referring to STED data? If so, this should be stated in the text.

14. I don't understand the labeling of the blue and grey bars in Figure S5J.

15. In Figure S6E, it may be easier for the reader to reach a conclusion with the single color images also shown separately.

16. Arrows in the supplementary movies 2 and 3 would help the reader see examples of colocalization.

Referee #2:

This manuscript presents the discovery of an atypical RNA-binding tropomyosin, DmTropomyosin1-I/C, that recruits kinesin to oskar RNP particles to mediate their transport to the posterior pole of the *Drosophila* oocyte. Although kinesin-dependent transport of oskar is a crucial step in establishment of embryonic polarity and the germline, how the oskar RNP is connected to kinesin has been a mystery. This paper answers that question. By identifying a new role for a tropomyosin as a RNA-kinesin adaptor, this work also resolves a longstanding question beginning with the original identification in 1995 of mutations in DmTmII that disrupt oskar localization, about what a tropomyosin might have to do with oskar transport. It also elucidates an unexpected pathway by which kinesin becomes connected to oskar, through the recruitment of kinesin to the nurse cell nuclear envelope. Finally, it nicely ties together the roles of the SOLE and the oskar 3'UTR in directing transport of oskar RNPs.

I have no substantive criticisms of the paper, given that the authors have gone to great lengths to address prior critiques. Although a paper on this topic was recently published, that study presented the view from 1000 feet, while this one presents the view from 200 nm. The level of depth here certainly warrants publication in EMBO J (although in fairness to the prior publication, it should be cited in this paper or at the least, mentioned in a note at the end).

Several minor issues should be addressed:

- 1) The first sentence of the introduction is missing a period.
- 2) The third sentence of the introduction has a misplaced comma (after mRNAs) that makes it difficult to read.
- 3) The wide swath of EB1 in Figure 1C is hard to parse - what part should we be focusing on? Shouldn't EB1 trajectories be apparent? Maybe some explanation in the legend would help.
- 4) MCP used in the experiment shown in Figure 2C is not explained. Is this MCP attached to osk RNA (but osk appears to be labeled by MCP-mcherry) or to something else? What is it measuring? Also the authors should explain that Staufen is a component of the oskar RNP. I could not find anything in the text that introduced the reason this protein was used here and for readers who aren't in the know, it would be helpful.
- 5) The authors state that "Although STED superresolution microscopy further increased crowding by resolving the confocal objects (Fig S5C, C' and G), it confirmed that both Khc-EGFP or EmGFP-Tm1-I are recruited to oskar mRNPs (Fig S5D). I couldn't figure out how the data in C and C' support the numbers in D. The legend states that autofluorescence is shown in magenta and oskar in

green, so it isn't clear where the EmGFP-Tm1-I is shown. Is it possible that autofluorescence is a mistake and that the authors mean direct GFP fluorescence from Em-GFP-Tm1-I? But in that case, there doesn't appear to be any overlap between that and oskar mRNA in the confocal image.

6) This reviewer was thoroughly confused by Figure S1E-H. As in point #4, what is MCP? The legend states: "colocalization of oskMS2-mCherry mRNPs with different GFP fusion proteins". What MCP-GFP is colocalizing with oskMS2-mCherry?

1st Revision - authors' response

28 November 2016

We thank the referees their constructive comments and suggestions on our work describing the critical role of atypical Tomopmyosin1-I/C in recruiting Kinesin heavy chain to transported oskar mRNPs.

Referee #1:

Kinesin motors play an important role in transporting mRNAs within the cytoplasm in many cell types. How mRNAs are linked to kinesins is, however, poorly understood. The manuscript provides evidence that a product of the tropomyosin locus (TM1-I/C) provides a long sought after link between oskar mRNA and the kinesin-1 motor during *Drosophila* oogenesis. The importance of this work extends beyond the mRNA localization field to the motor field; data is presented that supports a novel mechanism in which kinesin-1 is bound to cargo in an inactive form to be activated at a specific stage. The authors propose that this is a mechanism that minimizes interference with dynein motors that transport oskar mRNA from the nurse cells to the oocyte (until mid-oogenesis when kinesin-1-based transport is activated).

Several of the findings (e.g. on RNA binding activity and motor inhibition/activation) go beyond what was addressed in the recently published paper by Gonsalvez and colleagues that also proposed that the TM1 isoform is a link between oskar mRNA and kinesin-1 (note that my review takes into consideration that the Gaspar paper has been granted scoop protection by EMBO Journal). The data examining colocalization of TM1-I/C, Khc and oskar mRNA in timelapse movies of ex vivo transport assays, which appear to have been produced in response to the comments of the previous reviewers, make a more convincing case for some of the proposed functions of TM1-I/C than the data in Figure 2. However, the manuscript is still difficult to follow in places and there are still some significant issues that should be addressed by changes to the text, figures and movies.

Major points:

1. The authors do a commendable job of controlling for false positives in their analysis of colocalization data. I am less convinced that they are able to control for false negatives, i.e. complex formation being underestimated due to high levels of free proteins in the cytoplasm and/or some complexes containing a relatively low copy number of the fluorescently labeled protein. The data in Figure 3A and B, for example, appear to illustrate the difficulties in dismissing binding of one component to a motile complex containing oskar mRNA (due to high cytoplasmic protein signal). This is significant as some of the conclusions are based on the absolute colocalization frequencies (the conclusions examining relative differences in different genotypes are, of course, not affected).

We agree with the referee that the high cytoplasmic protein signal indeed renders our analyses difficult as it increases the false positive detection rate, and we therefore primarily aimed to control this in our analyses pipelines. However, our measurements show that ~50% of motile oskar mRNPs are in stable complex with Khc – which is close to the 65% of plus-ended runs of oskar particles, not all of which are mediated by Khc. Therefore we conclude that the colocalization frequencies of Khc and oskar that we measured in time-lapse images are close to being absolute. In the text we now address directly the question of the false-negative detection.

Page 6, line -10: " This revealed that in KhcmKate2 homozygous ooplasmic extracts nearly 50% of motile oskMS2-EGFP mRNPs are associated with Khc during at least half of the recorded trajectories (Fig 3C). As this value is close to the proportion of plus-end directed runs (65%) – not all of which are mediated by Khc - (Fig 1D), and the fraction of Khc positive mRNPs is proportional to the amount of labelled Khc (Appendix Figure S1G), we assume that the rate of

false negative detection is low in this analysis. In contrast to the high degree of association of Khc to motile mRNPs, only ~15% of non-motile oskar particles were found to be in complex with Khc during their trajectories in the same analysis.”

To give an example, the authors use the value of 15% of oskar mRNPs having detectable enrichment of Khc to argue for a model in which kinesin-1 dynamically turns over on the mRNP population. Do the authors have data that provides a convincing case that Khc is not bound to oskar mRNPs much more frequently (for example after an initial recruitment process at the nuclear envelope), and that colocalization can only be detected above background in 15% of cases? I think that this alternative model should be discussed or data presented/discussed that argues against it. At the moment the authors state in the abstract that a small but dynamically changing subset of oskar mRNPs gets loaded with inactive kinesin-1 and this statement is not fully justified by the current manuscript in my opinion (see next point).

We thank the referee for this criticism of our data. Unfortunately, we don't have any other means to look at Khc recruitment in the nurse cells than the snapshot analysis of fixed specimens. As we found that the levels of Khc positive mRNPs is not different in wild-type nurse cells and wild-type oocytes in such an analysis (Fig 5), we think that the true level of Khc association is not considerably higher in the nurse cells than in the oocyte. Even if it were, the biological importance of this Khc loading manifests in the oocyte during mid-oogenesis, when our assays almost unambiguously (see previous point) show that only a minority (~15%) of oskar mRNPs is in complex with Khc

2. The data that (the authors claim) backs up the dynamic association model by purporting to show association of Khc with mRNPs that precedes transport (Figure S3B) is weak. To back up this claim it would be necessary to show data from timepoints even earlier than those shown in Figure S3B, when one might expect Khc levels to be lower than on the transported mRNP. Even if that is the case, it is not clear that the levels of enrichment are significant, especially as a similar level of enrichment is seen for TM1-I/C. By the authors' own admission, the number of events that can be scored limits the power of this analysis. Unless this analysis can be strengthened by more data it should be omitted from the manuscript. This section of the manuscript would also be stronger if several example images of dynamic association of Khc with oskar mRNPs (quantified in Figure 3F) were provided in the supplementary section.

We agree with the reviewer that the results shown previously in Figure S3B - and consequently the derived conclusions on the delay between Khc recruitment and the onset of oskar motility - are rather weak due to the low sample size of observations on the past of motile oskar mRNPs (especially prior to three seconds from the start of the run). Therefore we decided to remove this analysis from our manuscript, as it does not allow us to draw a strong conclusion.

In contrast, the analysis of dynamic association of Khc (presented now in Fig 3F and G) is firm and shows that 1) Khc loading to oskar mRNPs is dynamic and it occurs on a time scale of seconds and 2) this dynamic association depends on the presence of Tm1-I/C. The revised manuscript now includes an example kymograph showing such Khc association events to a motile oskar mRNP (Fig 3F).

3. Why are mRNPs that colocalize with TM1-I/C no more likely to exhibit motile behaviour than non-motile behaviour (Figure 3D)? The authors should consider offering an explanation/hypothesis for this observation. Could it be that TM1-I/C is a constitutive component of the mRNP (necessary for Khc binding) but that scoring colocalization is difficult because of background levels of the protein in the cytoplasm?

Indeed, we have also been puzzled by our finding of the relatively low association of Tm1-I/C even, with motile oskar mRNPs. We think that – in contrast to Khc-mKate2 – the co-localization analysis between oskar and mCherry-Tm1-I suffers from a rather high false negative detection rate, as indicated by the non-proportional dependence of Tm1-I positive oskar mRNPs on mCherry-Tm1-I levels. The fluorescence of mCherry-Tm1-I was rather low compared to Khc-mKate2, requiring longer exposures during image acquisition to obtain similar fluorescence. Such technical difficulties could result in our inability to score a great portion of fluorescently labelled Tm1-I in the immediate vicinity of oskar mRNPs. These issues are now discussed in the manuscript:

Page 7, line 3:” We also found that approximately 20% of oskar mRNPs are stably associated with mCherry-Tm1-I/C irrespective of their motility (Fig 3B and D). The observed low and non-linearly scaling proportions of oskar mRNPs (Fig 3D) associated with the relatively dim

mCherry-Tm1-I (Figure 3B) suggest that (in contrast to Khc-mKate2, see above) the false negative detection rate in this analysis is rather high. Therefore we cannot reliably determine the true extent of the association and whether one fifth or a greater proportion of oskar mRNPs are associated with Tm1-I/C. Nevertheless, our data show that, like Staufen (Fig 3E), Tm1-I/C is a component of oskar mRNPs in the oocyte."

Figure 3B legend: "Note that mCherry-Tm1-I was exposed twice as long as Khc-mKate to obtain comparable red fluorescence signals"

4. I agree with Reviewer 1 that calling TM1-I/C a tropomyosin or an atypical tropomyosin could cause some readers confusion. Does this protein encode a coiled coil, which is a key feature of tropomyosins? Even if it does, it may be best to be conservative and refer to the protein as a product of the tropomyosin locus in the title and abstract, as many readers won't look into the manuscript for details.

We ourselves have dwelled on this issue at length. Although without structural evidence we cannot comment on the structure of the protein - as most of it is composed of low complexity sequences-, the fact is that it contains a short C-terminal, conserved tropomyosin superfamily domain. More importantly, this same isoform was shown to bind polymerized actin in vitro by the Montell lab in 2011. Also, the two recent papers from the Montell and the Gonsalvez labs refer to the Tm1-I/C isoform as "atypical tropomyosin" and "novel isoform of ... non-muscle Tropomyosin", respectively. For these reasons and to avoid confusion, we think that Tm1-I/C should be referred to as "atypical tropomyosin"

5. Figure S4A. Were these data derived from the same experiments presented in the text with the longer 3'UTR fragment? If so, the data for the longer 3'UTR should be shown for comparison in Figure S4A. If not, it seems that concluding that the 3'UTR truncations compromise RNA binding is not justified.

We agree with the reviewer. Since that particular experiment shown in Figure S4A (currently Fig EV3A) included no full length 3' UTR as positive control, we now show results of another experiment where Tm1-I/C binding of both of the truncations (UTR1-759 and UTR371-961) were tested against the full length 3' UTR under cross-linking conditions (Fig EV3C).

6. The Biorxiv version of the paper includes an additional way of presenting the results in Figure 7 of the current manuscript (Figure 7L in the preprint), which is very helpful for understanding the dataset. I recommend that this is reinserted in the manuscript. The preprint also has a clearer description of the advantage of tagging endogenous proteins when evaluating colocalization.

In the revised manuscript, we have integrated Figure 7L from the first preprint version of the manuscript; it now appears as Fig 7F.

Minor points:

1. I also agree with the reviewer's comment that some information on the features of the TM1-I/C protein (relationship to tropomyosin/presence of low complexity regions) should be introduced in the Results so that the reader can keep this in mind when reading the rest of the manuscript. I suggest that the authors consider this issue again.

We have included a clause on the domain structure of Tm1-I/C in the last paragraph of the Introduction:

Page 4, line -8: "Here, we demonstrate that DmTm1-I/C, which consists mainly of low complexity sequences and a C-terminal short tropomyosin superfamily domain (Cho et al., 2016),..."

2. It wasn't clear to me if the authors currently think that TM1-I/C is the only recruiter of Khc, if there could be an independent mechanism for recruiting some Khc, or if this question needs more work to resolve in the future (e.g. there is a subheading at the beginning of the Results stating that TM1-I/C 'maintains' kinesin-1 on oskar mRNAs).

Although we tried to indicate in the results and in the discussion that we observe a dependence on Tm1-I/C in case of the majority of - but not of all - Khc loaded onto oskar mRNPs, the subheading indeed suggested the opposite, as pointed out by the referee.

We have changed the subheading to "Tm1-I/C maintains proper levels of kinesin-1 on oskar mRNA"

Also, we now discuss the remainder of Khc associated with oskar mRNPs, in the Discussion:

Page 13, line 2: “It is also apparent from our data that a smaller amount of Khc is recruited to oskar mRNPs independent of Tm1-I/C. However, the dynamic exchange of these Khc molecules appears to be rather slow and, most likely as a consequence, they mediate only a minuscule fraction of intra-ooplasmic oskar transport – as inferred from the almost complete loss of motility in Tm1gs mutants (Appendix Table S1) – highlighting the importance of Tm1-I/C in kinesin loading.”

3. Tm1-I/C 'acts upstream' of Khc. This genetic term should be avoided when talking about a biochemical complex.

We indeed used the wrong terminology when writing “Tm1 acts upstream of Khc”. The correct sentence should have read as “Tm1 acts upstream of Khc”, especially as at this point of the manuscript we haven't yet identified Tm1-I/C. To avoid any confusion, we have removed the clause.

4. Figure 2A and B. The data would presumably be clearer if the single color images are shown in this figure or a supplementary figure.

We now added grayscale versions of the individual channels and also replaced the original merged images which as presented were moderately overexposed.

5. "Although such an en mass co-IP analysis lacks a spatiotemporal resolution, it shows that Tm1-I/C, kinesin-1 and Staufen indeed form complexes...". The analysis shows that Tm1-I/C forms complexes with kinesin-1 and that it forms complexes with Staufen. Although it is likely to be the case, the data do not show that all three components are in the same complex. The authors should consider modifying their language to make this clearer.

Indeed, the referee is right with their interpretation. We rephrased this sentence and now it reads as :

Page 8, line 11: “Although this en mass co-immunoprecipitation analysis cannot resolve either the heterogeneity or the spatiotemporal distribution of such complexes, it shows that Tm1-I/C forms complexes both with kinesin-1 and with Staufen that are very likely maintained by – not necessarily direct - protein-protein interactions.”

6. Figure 6 title. 'Transient' is not necessary as the figure does not address the persistence of the oskar association at this site.

We removed the word “Transient”.

7. The nature of oskar delta I (1,2,3), and its relationship to the SOLE/EJC, needs to be explained when first introduced. This is essential for understanding the logic of this very informative experiment.

In the Introduction, we now describe precisely the nature of oskar $\Delta i(1,2,3)$:

Page 3, line -1: “This second step of oskar transport requires splicing of the first intron in the oskar pre-mRNA, as oskar transcripts lacking all three introns (oskar $\Delta i(1,2,3)$) or just intron 1 (oskar $\Delta i(1)$) fail to localize (Hachet & Ephrussi, 2004). This splicing event results in assembly of the spliced localization element (SOLE) and deposition of the exon junction complex (EJC) on the mRNA (Ghosh et al., 2012).”

8. Figure S7C and D should be called in the text when referring to the enrichment of Egl and p50 around the nuclear envelope.

We thank the referee for pointing out the absence of these two references,. We now refer to these figures as Appendix Fig S2C and Appendix S2D, respectively, at the appropriate places.

9. References need to be provided in the methods for the pan-TM1 and Egl antibodies.

The two references are now provided.

10. Figure S4B. What is FMG?

FMG is now spelled out as FlagMycGFP.

11. Figure S4C and D legend. The text does not match the panels (rather it describes data in Figure S5). I therefore couldn't deduce what S4C and D show.

We thank the referee for noticing this and apologize for the confusion. There was a vestige of the legend of Figure S5 in the legend of Figure S4. It has now been removed.

12. I don't understand what Figure S5C and C' are supposed to show as not much information is given about this experiment in the text. Also, in the corresponding legend, 'auto-fluorescence' is referred to for GFP. Presumably 'fluorescence' could be used to prevent potential confusion with cytoplasmic autofluorescence?

We have rewritten the sentence as follows:

Fig EV4C (previously Figure S5) legend: "EmGFP-Tm1-I is in magenta, oskar mRNA is in green."

13. Is Figure S5D referring to STED data? If so, this should be stated in the text.

For greater clarity, we have added "in STED images" to the end of the sentence in the legend to Fig EV4D (previously Figure S5).

14. I don't understand the labeling of the blue and grey bars in Figure S5J.

We now included reference to the colours of the bars in the figure legend:

Legend to Fig EV4J (previously Figure S5): "Mean density of the detected Khc-mKate2 (blue) and mRNPs labelled by the oskar 3' UTR probe set (gray) in the nurse cells"

15. In Figure S6E, it may be easier for the reader to reach a conclusion with the single color images also shown separately.

In Fig EV5E (previously Figure S6), we now additionally show both colours as separate grayscale images in panels E' and E".

16. Arrows in the supplementary movies 2 and 3 would help the reader see examples of colocalization.

We agree with the referee and apologize for submitting the non-annotated versions of these movies. We have added the arrows (which were present in the original preprint version of the manuscript)..

Referee #2:

This manuscript presents the discovery of an atypical RNA-binding tropomyosin, DmTropomyosin1-I/C, that recruits kinesin to oskar RNP particles to mediate their transport to the posterior pole of the *Drosophila* oocyte. Although kinesin-dependent transport of oskar is a crucial step in establishment of embryonic polarity and the germline, how the oskar RNP is connected to kinesin has been a mystery. This paper answers that question. By identifying a new role for a tropomyosin as a RNA-kinesin adaptor, this work also resolves a longstanding question beginning with the original identification in 1995 of mutations in DmTmII that disrupt oskar localization, about what a tropomyosin might have to do with oskar transport. It also elucidates an unexpected pathway by which kinesin becomes connected to oskar, through the recruitment of kinesin to the nurse cell nuclear envelope. Finally, it nicely ties together the roles of the SOLE and the oskar 3'UTR in directing transport of oskar RNPs.

I have no substantive criticisms of the paper, given that the authors have gone to great lengths to address prior critiques. Although a paper on this topic was recently published, that study presented the view from 1000 feet, while this one presents the view from 200 nm. The level of depth here certainly warrants publication in EMBO J (although in fairness to the prior publication, it should be cited in this paper or at the least, mentioned in a note at the end).

We completely agree with the referee that any relevant previous studies, regardless how recently they were published, should be properly discussed and referenced. In fact we had included the citation and discussion of the Veeranan-Karmegam et al., 2016 paper in the previous version in the manuscript, nevertheless, we now include more of their results in the Discussion.

Page 12, line 10: "Here, we have shown that the majority of kinesin-1 motor associated with oskar mRNA is recruited by Tropomyosin1-I/C, a non-canonical RNA binding protein, which explains the mislocalization of oskar mRNA when Tm1-I/C is lacking (Erdelyi et al., 1995;

Veeranan-Karmegam et al., 2016) (and this study)."

Several minor issues should be addressed:

- 1) The first sentence of the introduction is missing a period.
- 2) The third sentence of the introduction has a misplaced comma (after mRNAs) that makes it difficult to read.

We thank the referee for spotting these punctuation errors, both of which are now corrected.

- 3) The wide swath of EB1 in Figure 1C is hard to parse - what part should we be focusing on? Shouldn't EB1 trajectories be apparent? Maybe some explanation in the legend would help.

We have now highlighted the growing plus tip of the MT (decorated with EB1-mCherry) with a white dashed arrow.

Figure 1C legend: "White dashed arrow shows the growing plus tip of the MT"

- 4) MCP used in the experiment shown in Figure 2C is not explained. Is this MCP attached to osk RNA (but osk appears to be labeled by MCP-mcherry) or to something else? What is it measuring? Also the authors should explain that Staufen is a component of the oskar RNP. I could not find anything in the text that introduced the reason this protein was used here and for readers who aren't in the know, it would be helpful.

We apologize for the confusion caused by our use of 'MCP' in the figure, with no further explanation., We have now added an explanatory sentence to the figure legends:

Figure 2C legend (and EV1 legend): "MCP indicates MCP-EGFP which, like MCP-mCherry, can bind to MS2 loops. Staufen (Stau) is a dsRNA binding protein and bona fide partner of oskar mRNA (St Johnston et al., 1991; St Johnston et al., 1992)."

- 5) The authors state that "Although STED superresolution microscopy further increased crowding by resolving the confocal objects (Fig S5C, C' and G), it confirmed that both Khc-EGFP or EmGFP-Tm1-I are recruited to oskar mRNPs (Fig S5D). I couldn't figure out how the data in C and C' support the numbers in D. The legend states that autofluorescence is shown in magenta and oskar in green, so it isn't clear where the EmGFP-Tm1-I is shown. Is it possible that autofluorescence is a mistake and that the authors mean direct GFP fluorescence from Em-GFP-Tm1-I? But in that case, there doesn't appear to be any overlap between that and oskar mRNA in the confocal image.

We removed the confusing term of GFP-autofluorescence and replaced the sentence with the following:

Fig EV4C (previously Figure S5) legend: "EmGFP-Tm1-I is in magenta, oskar mRNA is in green."

Indeed, these overlaps are rather rare – as also indicated by our analysis of tens of thousands of mRNPs (e.g. Fig EV4D and E).

- 6) This reviewer was thoroughly confused by Figure S1E-H. As in point #4, what is MCP? The legend states: "colocalization of oskMS2-mCherry mRNPs with different GFP fusion proteins". What MCP-GFP is colocalizing with oskMS2-mCherry?

We apologize for the lack of clarity and, as for point #4, we have added the explanatory sentence to the legend of Fig EV1E (previously Figure S1).

Figure 2C legend (and EV1 legend): "MCP indicates MCP-EGFP which, like MCP-mCherry, can bind to MS2 loops."

Corresponding Author Name: Imre Gaspar and Anne Ephrussi

Manuscript Number: EMBOJ-2016-96038